

# Soot aerosol from commercial aviation engines are poor ice nucleating particles at cirrus cloud temperatures

Baptiste Testa[1], Lukas Durdina[2], Peter A. Alpert[3], Fabian Mahrt[3], Christopher H. Dreimol[4,5],
Jacinta Edebeli[2], Curdin Spirig[2], Zachary C.J. Decker[3], Julien Anet[2], and Zamin A. Kanji[1]

[1]Institute for Atmospheric and Climate Science, ETH Zürich, Zürich, Switzerland
[2]Centre for Aviation, ZHAW School of Engineering, Winterthur, Switzerland
[3]Laboratory of Atmospheric Chemistry, Paul Scherrer Institute, Villigen PSI, Switzerland
[4]Wood Materials Science, Institute for Building Materials, ETH Zürich, Zürich, Switzerland
[5]Cellulose & Wood Materials Laboratory, Empa, Dübendorf, Switzerland

**Correspondence:** Baptiste Testa (baptiste.testa@env.ethz.ch) and Zamin A. Kanji (zamin.kanji@env.ethz.ch)

**Abstract.** Ice nucleating particles catalyze ice formation in clouds, affecting climate through radiative forcing from aerosol-cloud interactions. Aviation directly emits particles into the upper troposphere where ice formation conditions are favorable. Previous studies have used proxies of aviation soot to estimate their ice nucleation activity, however the investigations with commercial aircraft soot from modern in-use aircraft engine have not been quantified. In this work, we sample aviation soot particles at ground level from different commercial aircraft engines to test their ice nucleation ability at temperatures $\leq 228$ K, as a function of engine thrust and soot particle size. Additionally soot particles were catalytically stripped to reveal the impact of mixing state on their ice nucleation ability. Particle physical and chemical properties were further characterized and related to the ice nucleation properties. The results show that aviation soot nucleates ice at or above relative humidity conditions required for homogeneous freezing of solution droplets ($RH_{hom}$). We attribute this to a mesopore paucity inhibiting pore condensation and the sulfur content which suppresses freezing. Only large soot aggregates (400 nm) emitted under 30-100 % thrust conditions for a subset of engines (2/10) nucleate ice via pore condensation and freezing. For those specific engines, the presence of hydrophilic chemical groups facilitates the nucleation. Aviation soot emitted at thrust $\geq$100 % (sea level thrust) nucleates ice at or above $RH_{hom}$. Overall our results suggest that aviation soot will not contribute to natural cirrus formation and can be used in models to update impacts of soot-cirrus clouds.

## 1 Introduction

While only about 10 % of the global population fly (Gössling and Humpe, 2020), the aviation sector contributes about 3.5 % of the total net anthropogenic effective radiative forcing (2300 $mWm^{-2}$; Lee et al., 2021) warming the Earth-atmosphere system. Non-$CO_2$ emissions account for two thirds of the warming component. The current understanding of radiative forcing from aviation soot aerosol-cloud interactions is very low and from a recent study, no best estimates were provided due to a lack of agreement on magnitude and sign (Lee et al., 2021). In fact, the radiative forcing associated with aircraft soot aerosol-cloud interactions has been reported to range from -500 to +300 $mWm^{-2}$ (Zhou and Penner, 2014), although recent studies estimated



smaller effects (i.e., $< 50 \, \mathrm{mWm^{-2}}$; Zhu et al., 2022; Righi et al., 2021; Lee et al., 2021, and references therein).

The uncertainties on the radiative forcing estimates arise from the poor understanding and diverse assumptions about aviation soot perturbing background cirrus by acting as ice nucleating particles (INPs) in global climate models (Lee et al., 2021; Righi et al., 2021). Aviation soot aerosol particles are mainly emitted at cirrus altitudes ($\sim 11 \, \mathrm{km}$), where the background aerosol loading is low (e.g., mineral dust number concentration is $10 - 20$ times less than aviation soot number concentration at flight altitude; Righi et al., 2021). Depending on the ability of aviation soot to serve as INPs, their capacity to consume the available water vapor through ice formation and growth potentially affects cirrus clouds in those regions.

The ice nucleation abilities of aviation soot particles that are directly emitted from aircraft engines currently in use has not been quantified, as such, proxies of these soot particles from laboratory studies have been used to represent aviation soot ice nucleation abilities. A large body of work has investigated the ice nucleation abilities of different laboratory generated soot types. Results of these studies have been summarized elsewhere (e.g., Marcolli et al., 2021; Kanji et al., 2017; Ullrich et al., 2017). While soot was found not to be ice active at temperatures $T > 235 \, \mathrm{K}$ (Kanji et al., 2020), some soot types were potent INPs at temperatures and relative humidity (RH) below that required for the homogeneous freezing of solution droplets ($T_{\mathrm{hom}}$ and $\mathrm{RH_{hom}}$, respectively) (Koehler et al., 2009; Mahrt et al., 2018; Nichman et al., 2019; Zhang et al., 2020a; Mahrt et al., 2020b, a; Gao and Kanji, 2022; Gao et al., 2022b, c). It is now well established that at cirrus temperatures and $\mathrm{RH} < \mathrm{RH_{hom}}$ soot particles mostly nucleate ice via pore condensation and freezing (PCF), (Mahrt et al., 2018; Nichman et al., 2019; Zhang et al., 2020a; Marcolli et al., 2021; Gao et al., 2022a, b).

PCF is a three-step mechanism (Marcolli, 2014; Campbell and Christenson, 2018; David et al., 2019; Jantsch and Koop, 2021):
In a first step, water vapor condenses in soot aggregate pores (e.g., cavities or voids) below bulk water saturation $\mathrm{RH_w} < 100 \, \%$ which is followed by pore water freezing homogeneously at $T < T_{\mathrm{hom}}$. In the third step, pore ice grows out of the pore into a macroscopic ice crystal by diffusional growth. For porous particles, the required water vapor pressure for pore filling decreases with pore diameter and contact angle between the pore and the water surface (Marcolli, 2014, 2020). Moreover, homogeneous ice nucleation of pore water at $T < T_{\mathrm{hom}}$ limits PCF to pores that can accommodate the critical ice cluster (David et al., 2020). PCF in turn can occur only for pore of diameter between 2 and $50 \, \mathrm{nm}$ (Marcolli et al., 2021) that are termed mesopores (Haul, 1982). Porosity of the soot aggregates depend on the diameter of the primary soot particles ($D_{\mathrm{pp}}$), their spatial arrangement and their degree of overlap (Brasil et al., 1999; Marcolli et al., 2021). Previous studies on laboratory soot have found aggregate size (Mahrt et al., 2018; Gao et al., 2022a) and surface chemical groups to impact ice nucleation via PCF (Zhang et al., 2020b; Mahrt et al., 2020a; Gao et al., 2022b) by impacting pore volume and soot-water contact angle (hydrophilicity). These previous studies have been invaluable for our understanding of soot ice nucleation, but much less is known about the ice nucleation of soot particles emitted from real jet engines which could have considerably different physicochemical properties than the surrogates.

Aircraft soot particle composition and physical properties strongly depend on many factors including engine type, engine thrust and fuel type (Vander Wal et al., 2010; Atiku et al., 2016; Durdina et al., 2021a). As an example, sulfur compounds present in aviation fuel (about 500-600 $\mathrm{ppm}$ for Jet A-1 fuel; Starik, 2007; Lee et al., 2021) can be internally mixed with aviation soot (Schumann et al., 2002; Parent et al., 2016). Sulfur, in the form of $H_2SO_4$ is known to affect the particle water uptake



by facilitating water absorption (Gysel et al., 2003; Popovicheva et al., 2004; Demirdjian et al., 2009). However, depending on the amount, $H_2SO_4$ coverage can inhibit ice nucleation in soot aggregate mesopores via PCF by lowering the water activity and nucleation rate of ice (Gao et al., 2022c). Similarly, different engine types and models with different combustor designs emit

aerosols with different properties (e.g., effective density; Durdina et al., 2021a). This suggests that the agglomerates porosity might vary between engine types, affecting the ice nucleation ability of the soot particles. Aviation soot particle morphology and chemical content will change depending on the engine thrust (Vander Wal et al., 2022). The mode diameter of soot aggregates generally increases from < 10 nm at idle (7 % thrust) up to around 50 nm at higher thrust (Durdina et al., 2021b; Liati et al., 2019). The number of primary particles generally increases with aggregate size (Sorensen, 2011). Similarly, averaged

aggregate $D_{pp}$ has been shown to increase with thrust, potentially resulting in different soot porosity at different engine thrust. The engine thrust also affects the particle crystallinity and organic carbon content (Vander Wal et al., 2014; Liati et al., 2014; Abegglen et al., 2016; Delhaye et al., 2017; Marhaba et al., 2019; Liati et al., 2019). This is known to control the particle water uptake capacity (Haul, 1982; Persiantseva et al., 2004; Popovicheva et al., 2011). The above characteristics mean that the ice nucleating ability can be expected to be influenced by thrust level.

Overall, real aircraft soot particles can have vastly different properties to those of laboratory generated soot types. It therefore remains unclear to what degree previous ice nucleation measurements based on proxies of aircraft soot particles can be used to predict the impact of aviation soot on cirrus clouds. To address this uncertainty, we quantified the ice nucleation abilities of soot particles emitted from in-use commercial aircraft turbines at $T < 228$ K. The influence of engine thrust on the ice nucleation is investigated as well as the dependence on key particle physical and chemical properties, including aggregate size, morphology,

chemical composition and mixing state.

## 2    Method

### 2.1    Engine emission tests and exhaust sampling

Soot particles were sampled from commercial aircraft engines at the maintenance facility, SR Technics at Zürich airport. Sampling took place from several engine from P&W and CFM International, all fueled with Jet-A1 fuel. A detailed description

of the sampling system at the engine test-cell facility is given elsewhere (Durdina et al., 2014; Abegglen et al., 2016; Lobo et al., 2020). In brief, the aircraft engine exhaust was sampled with a single-orifice heat-resistant alloy probe positioned ∼1 m downstream of the engine exhaust nozzle in the test-cell (Fig. 1). The engines were operated following a standardized procedure, starting at ground idle (7 % sea level thrust) and increasing step-wise from low thrust (about 20 % sea level thrust) to high-thrust corresponding to take-off (∼ 100 % sea level thrust), stabilizing at 6 different thrust points for few minutes.

Idle emissions are ground-based and are relevant only in the airport and surrounding area. Therefore, the engine exhaust was first sampled after the engine reached the first thrust step above ground idle (about 20 % sea level thrust) and until the maximum thrust point, resulting in a total of about 15 min of emissions sampling (Fig. A1a). In the following, we refer to those experiments as "mixed-thrust" experiments. To asses possible effect of the thrust on the ice nucleation abilities of aviation soot, we targeted a second engine experiment procedure ("high-thrust" experiment) where the engines run only at high thrust for





about 5-7 min (Fig. A1b). A list of models and power settings of the engines tested is shown in Table B1. Since the time required for the ice nucleation measurements (several hours) was longer than the soot emission time (5 to 15 min), the exhaust was collected into an aerosol tank, a $0.3 \ \mathrm{m}^3$ stainless steel reservoir, from which particles were sampled for ice nucleation measurements. Transport from the engine test-cell to the instrument room took place via an insulated trace-heated (433 K, 12 m length and 10 mm inner diameter) tubing, without dilution. The tank was kept at room temperature and was continuously stirred to keep the particles suspended. To prevent particle restructuring due to water vapor uptake while cooling the exhaust down to room temperature, the exhaust was dried ($\mathrm{RH_w} < 10 \ \%$) prior to entering the aerosol tank with a set of molecular sieve and silica gel driers connected in series (Fig. 1). This study does not focus on emission quantification, hence particle losses through the sampling lines were not assessed. Nevertheless, volatile compounds remaining downstream of the driers can condense onto the soot surface. The influence of the sampling system on the soot mixing state will be discussed in Sect. 4. The total sampling flow rate from the engine into the aerosol tank ranged from 10 to $30 \ \mathrm{L\,min^{-1}}$ depending on the pressure (i.e., engine thrust) at the probe inlet. Total number concentration of emitted soot particles at ranged from $1 \times 10^5$ to $1 \times 10^7$ $\mathrm{cm}^{-3}$, hence particle coagulation is expected within the first few minutes upon entering the restricted volume of the aerosol tank. After emission, the aerosol tank was isolated from the probe. The exhaust particles were kept suspended for at least 1h to allow particle coagulation to reduce. Coagulation is a function of particle concentration and size and led to shift of the mode diameter from 30 nm measured at the engine exit plane (Durdina et al., 2021b), to 70-450 nm (measured downstream of the aerosol tank, Fig. E1 panels a-e). After a coagulation time of 1 h, the tank was connected to the ice nucleation and aerosol instruments (Fig. 1).

## 2.2 Particle catalytic stripping and sizing

The soot particles sampled from the tank could be passed through a catalytic stripper (CS08, Catalytic Instruments, $T$ = 623 K, equipped with a sulfur trap) when desired. Coagulated particles in the aerosol tank without further processing are called unCS-soot and those that are catalytically stripped are referred to as CS-soot. The catalytic stripper desorbs sulfur and any semi-volatile material condensed on the soot, therefore changing its mixing state. When operated with air, volatilized organics at this high temperature can become oxidized at the walls of the stripper and irreversibly adsorbed with no chance to recondense onto the soot particles. Desorbed materials include organic carbon and sulfur compounds from the soot surface, cavities or pores. We note that the efficiency of removal of organic volatile compounds can degrade overtime due to sulfur depositing on the catalyst, reducing the efficiency of removal to below 99 %. A first set of engine runs was dedicated to quantify the ice nucleation abilities at 218 K and 223 K of size-selected particles of 200 nm and 400 nm. The particles were selected with a differential mobility analyzer (electrostatic classifier 3082, DMA 3081 column and a X-ray source, TSI Inc) with an aerosol to sheath flow ratio of 1:12 and 1:7 for 200 and 400 nm, respectively. A second set of engine runs was dedicated to quantify the ice nucleation abilities at 218 K and 228 K of polydisperse soot particles. The size distributions of the polydisperse experiments varied from engine to engine due to differences in soot emission index and coagulation in the aerosol tank (Fig. E1 panels a-e).



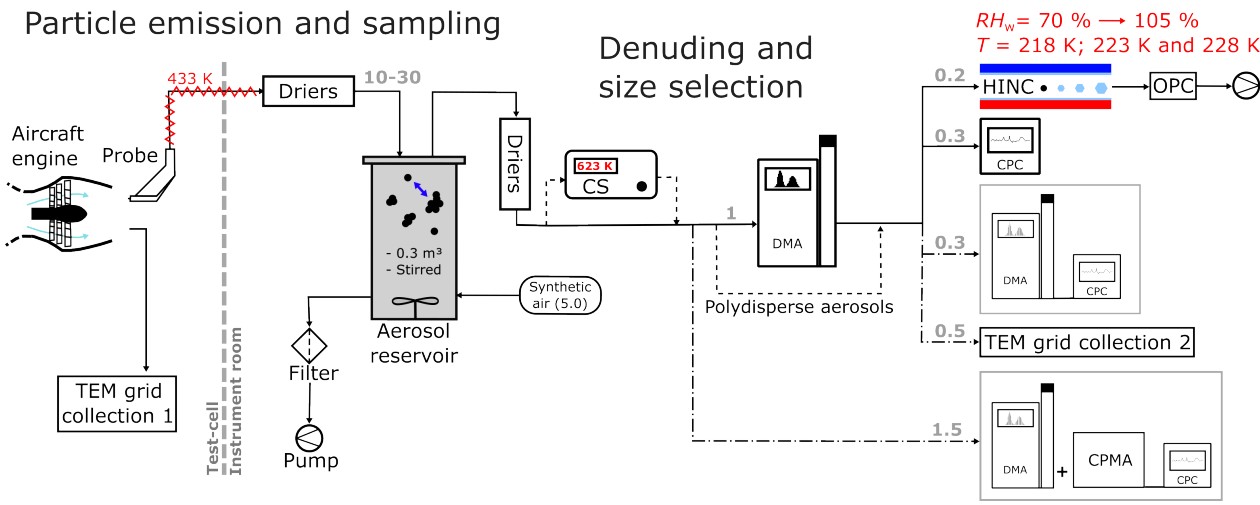

**Figure 1.** Experimental set-up used for the sampling of aviation soot for ice nucleation measurements and particle characterization. The arrows indicate the direction of the aerosol flow. The black dashed lines represent the aerosol path for the denuded and polydisperse experiments. Black dotted-dashed lines - mass measurements with the DMA + CPMA-CPC; particle collection onto TEM grids (TEM grid collection 2); and particle size measurement with the DMA-CPC - all not conducted simultaneously. Flow rates are indicated in grey and are in $\mathrm{L\,min^{-1}}$. TEM = transmission electron microscopy; CS = catalytic stripper; DMA = differential mobility analyser; OPC = optical particle counter; CPC = condensation particle counter; CPMA = centrifugal particle mass analyser; HINC = Horizontal Ice Nucleation Chamber.

## 2.3 Ice nucleation measurements

The ice nucleation measurements were conducted with the Horizontal Ice Nucleation Chamber (HINC). HINC is a continuous flow diffusion chamber (Rogers, 1988) which allow to quantify the ability of aerosol particles to nucleate ice as a function of RH and $T$. The working principle of HINC has been already documented (e.g., Lacher et al., 2017) and is briefly summarized here. Aerosol particles enter the chamber and have a residence time of $\sim 10$ s during which they experience ice saturation ($\mathrm{RH_i} > 100$ %) allowing them to nucleate ice. The RH experienced by the aerosols is controlled by the ice coated chamber wall temperatures and increases with increasing wall temperature difference. The relative humidity was scanned from $\sim 70$ % $\mathrm{RH_w}$ (= 117 % $\mathrm{RH_i}$ at 218 K) to $\sim 105$ % $\mathrm{RH_w}$ (= 175 % $\mathrm{RH_i}$ at 218 K) in HINC at a rate of 1.2 $\mathrm{RH_w\,min^{-1}}$ (= 2 $\mathrm{RH_i\,min^{-1}}$ at 218 K, Fig. 1). The sample flow containing the aerosols is sandwiched by a high purity nitrogen sheath flow, summing up to a total chamber flow rate of 2.83 $\mathrm{L\,min^{-1}}$. The sample to sheath flow ratio determines the RH range experienced by the particles within the aerosol-lamina of HINC (Mahrt et al., 2018). The ratio was set to 1:10-1:12 for all experiments, resulting in a maximum $\mathrm{RH_w}$ uncertainty at $T$ = 218 K of 3 % and 4.5 % across the sample flow lamina at $\mathrm{RH_{hom}}$ = 91 % and $\mathrm{RH_w}$ = 100 %, respectively. Particles that entered HINC were < 1 µm, and those that nucleated ice when exiting the chamber were counted with an optical particle counter (OPC GT-526S, MetOne) which detects particles with diameters $\geq 0.3$ µm. Temperatures



investigated in this study are below that required for homogeneous ice nucleation of pure micron-sized water droplets. As only submicron aerosols were introduced in the chamber, supermicron (> 1 μm) particles detected by the OPC were considered as ice. The fraction of aerosol that nucleate ice is called the activated fraction (AF) and is given as $AF = \frac{OPC_{counts > 1\,\mu m}}{CPC_{counts}}$. Uncertainty on AF arises from uncertainties of the CPC and OPC counts (10 % each) as well as from uncertainties in the HINC sample flow rate (5 %) resulting in propagated error of ± 15 % in the AF.

## 2.4 Sample collection

To characterize the physical and chemical soot properties, the outflow of the aerosol tank (10 to 30 L min$^{-1}$) was collected onto 0.4 μm porous polycarbonate filters (ISOPORE, Merck KGaA, Darmstadt, Germany). The soot deposited onto the filters during the high-thrust and mixed-thrust experiments was scraped and stored in separate glass bottles. This sampling required combining soot from different engine experiments (and hence models) because the soot mass collected from a single engine run was insufficient for the analysis. Soot particles were also collected onto transmission electron microscopy (TEM) grids (copper Formvar/Carbon, TED PELLA INC.) for microscopy and spectroscopy analysis downstream of the test cell (TEM grid collection 1 in Fig. 1). There, particles are rapidly diluted with the air flow passing through the engine and are thought not to undergo coagulation and morphological change before sampling. Particles were also collected downstream of the aerosol tank (TEM grid collection 2 in Fig. 1) as polydisperse and monodisperse unCS and CS samples, onto separate grids. Two particle collection apparatus were used which are the Zurich Electron Microscope Impactor (Mahrt et al., 2018), and the Partector TEM (Naneos Particle Solutions gmbh) Martin Fierz and Burtscher (2014).

## 2.5 Chemical and physical characterization of the samples

### 2.5.1 Electron microscopy

Particle morphology was investigated using TEM (JOEL-JEM 1400) operated at 120 kV. The resulting images were used for morphology analysis by image and processing with the MATLAB (R2020a, MathWorks Inc., Natick, USA) code of Dastanpour and Rogak (2014), which was modified in order to retrieve the equivalent spherical diameter, convexity and circularity of the soot particles, following previous studies (e.g., China et al., 2015; Mahrt et al., 2018). A Talos F200X G2 high resolution TEM (HRTEM, ThermoFischer), operated at 200 kV was used to obtain images at higher magnification (Fig. C1) that allowed to qualitatively determine primary particle properties (e.g., size, overlap).

In addition, X-ray spectroscopy (EDX) coupled to the HRTEM was used to obtain information on the chemical composition. In EDX, ionization of core-electrons in the sample resulting from interactions with the electron beam leaves holes in the atom inner shell. Such holes can be filled by transitioning electrons from higher energy levels, which are accompanied by X-ray emission, with energies characteristic of a particular chemical element. Focusing and scanning the electron beam on a single particle (scanning TEM mode = STEM) while collecting the X-rays allow to map its chemical composition. Quantification of the mapped elements has been conducted following the Cliff-Lorimer ratio technique (Williams and Carter, 2009). Carbon, oxygen, sulfur and nitrogen were consistently selected for quantification. We point out that sulfur might get vaporized under



the vacuum in the TEM. Thus, sulfur estimates from STEM-EDX represents lower limits. Other elements such as Sn, P and Al were sometimes detected in trace amount. Cu quantification was excluded since it mainly arises from the grid.

### 2.5.2 Scanning transmission X-ray microscopy and near edge X-ray absorption fine structure spectroscopy (STXM/NEXAFS)

Soft X-rays having energies 200-600 eV were used to acquire STXM/NEXAFS spectra to observe the chemical bonding of compounds in the sample. STXM/NEXAFS is based on the absorption of monochromatic X-rays by the sample causing a reduction in the incident photon intensity. Briefly, core-electrons that absorb X-ray photons can transition to unoccupied molecular orbital or become ionized. K-shell (i.e., 1s) electrons can transition to $\pi$ or $\sigma$ antibonding orbitals (1s $\rightarrow \pi^*$ and 1s $\rightarrow \sigma^*$) and are referred to as $\pi^*$ and $\sigma^*$ transitions, respectively (Stöhr 1992). The energy level of the unoccupied orbitals varies depending on the chemical bonding environment of the atom (Moffet et al., 2011). For instance, a graphite carbon atom in soot absorbs X-ray photons of 285.4 eV, while organic carbon atoms (e.g., bonded with oxygen) would absorb around 286-288 eV (Hopkins et al., 2007). Therefore, information on the chemical bonding (e.g., functional groups, structure) of the samples can be obtained by scanning the photon energy, resulting in an absorption spectrum that is specific to the given sample. For carbon or any targeted element, this ranges from the pre-edge energy where everything else than the targeted element is absorbing, to the post-edge energy, i.e., above the ionization threshold of the element. Different elements absorb in different energy ranges ($\sim$300 eV for carbon and $\sim$550 eV for oxygen) and the absorption further depends on the sample absorption cross section, $\sigma$ and thickness $d$ (Moffet et al., 2011). The optical density (OD) due to X-ray absorption observations follows Beer-Lambert's law: $OD = -\ln(\frac{I}{I_0}) = \sigma d$, with OD being dimensionless, $I$ being the transmitted photon intensity and $I_0$ being the incident intensity. The error in OD arises from photon counts and sample thickness.

The STXM/NEXAFS measurements were conducted at the PolLux endstation at the Swiss Light Source, Paul Scherrer Institute, Villigen, Switzerland (Raabe et al., 2008). Absorption spectra at the carbon and oxygen absorption K-edges were collected for selected soot samples. unCS- and CS-soot samples were investigated and their chemical functionalities were compared to identify potential changes due to stripping. Spatial resolution was below 100 nm (smaller than the soot sizes) with an energy resolution of 0.1 eV at the carbon and about 1.3-1.6 eV at the oxygen K-edges sufficient to identify any peak at both edges. Energy calibration was achieved by measuring absorption spectra at the carbon K-edge of polystyrene particles and comparing with the literature (Urquhart et al., 2000; Dhez et al., 2003). In order to compare particles with different thickness and background absorption, OD between 278 and 282 eV (pre-edge) were subtracted from the energy-corrected spectra and normalized to total carbon by dividing by the integrated OD between 305 and 320 eV (Takahama et al., 2010; Shakya et al., 2013). The absorption spectra were also used to calculate the level of soot graphitization, which can affect the reactivity toward oxygen (Liati et al., 2014) and hence affect the particle's water affinity (Persiantseva et al., 2004). It can be deduced from the peak height ratio, $R$, of the $\pi^*$ graphite C=C transition (285.4 eV) to 1s to $\sigma^*$ transition of saturated and unsaturated carbon (292 eV), i.e., $R = \frac{OD_{C=C}}{OD_{C-edge}}$ (Liati et al., 2019).





### 2.5.3 Dynamic water vapor sorption

Information on soot hydrophilicity and porosity can be obtained with gravimetric sorption techniques. Here, we used a dynamic vapor sorption device (DVS, Model Advantage ET 1, Surface Measurement Systems Ltd., London, UK) in which changes in the the bulk soot sample mass are continuously monitored for increasing (sorption) or decreasing (desorption) RH at a constant temperature of 298 K. The $RH_w$ was increased by changing the ratio of humidified and $N_2$ flow the sample was exposed to, while keeping the total flow rate constant at $0.2 \, \mathrm{L \, min^{-1}}$. In our experiments the $RH_w$ changed from 0 % to 40 % in steps of 10 %, followed by steps of 5 % up to 80 % and finally up to 98 % in steps of 3 % to capture water uptake due to the presence of mesopores in the sample (Haul, 1982). A mass change of less than $0.0005 \, \% \, \mathrm{min^{-1}}$ during 10 min was considered as steady-state and used as a criteria to move to the next RH. In case of a slow water uptake process, a time limit of 1000 min was set as second criteria to consider a sample to be in a steady state at that $RH_w$ level. Any initial sample moisture was desorbed by outgazing the sample with pure $N_2$ for 1400 min at 313 K prior the DVS analysis. The mixed-thrust soot bulk sample was divided in two samples of about 1-1.2 mg each, and one sample was used to measure its water uptake. The second sample was first heated to 623 K in a thermogravimetric analyzer (TGA Q50, TA Instruments) oven in an synthetic air atmosphere in order to mimic catalytic stripping of the soot particles and is referred to as CS. Following the TGA measurements, the stripped sample was immediately analyzed by the DVS. Uncertainty on the mass measurement are 0.1 µg.

### 2.5.4 Centrifugal particle mass analyzer

The centrifugal particle mass analyzer (CPMA; Cambustion Ltd., Cambridge, UK) selects aerosol particles according to their mass to charge ratio (e.g., Olfert and Collings, 2005) and is routinely used to measure the mass of soot particles (Durdina et al., 2014; Johnson et al., 2015). In this study, the mass of size selected unCS and CS aircraft soot particles was measured to obtain information on their fractal dimension. In fact, for fractal particles, the mass scales with the mobility diameter following a power-law relationship, commonly referred as mass-mobility relation: $m_p = C \, d_m^{D_{fm}}$ (Olfert and Rogak, 2019). Here $m_p$ is the particle mass in kg, $C$ a constant with unit $\mathrm{kg \, m^{-D_{fm}}}$ called the mass–mobility prefactor and $d_m$ the particle mobility diameter in m. The value of $D_{fm}$ gives information on the fractal nature of the particles, with $D_{fm} = 1$ for line-shaped particles and $D_{fm} = 3$ for spherical particles (Sorensen, 2011). $D_{fm}$ has been retrieved for soot particles from several engine types by charging and size selecting the coagulated unCS aircraft soot particles and measuring the mass distribution with the CPMA-CPC system (Ghazi and Olfert, 2013; Abegglen et al., 2015). Mass modes were retrieved from least square fit of the mass distribution using a log-normal distribution. Using this method, $D_{fm}$ can be retrieved from the mass-mobility relation.

## 3 Results and discussion

### 3.1 Overview of ice nucleation measurements

Ice nucleation experiments were conducted for a total of 14 engine emission tests (10 and 4 for monodisperse and polydisperse particles, respectively; see Table B1) and are summarized in Fig. 2. The figure shows the AF as a function of RH for 218 K,





K and 228 K for the individual ice nucleation experiments, as well as for the median of all measurements at the given conditions. The ice nucleation onset, corresponding to the conditions where AF = $10^{-3}$ = 0.1 % is reached, for the median of the unCS samples shows ice nucleation at or close to $RH_{hom}$ at all temperatures and sizes, including the polydisperse soot regardless of the mode size (80-200 nm, Fig. E1a-e). Above $RH_{hom}$, the AF steeply increases characterised by homogeneous

freezing reaching a plateau at around 30 %. Overall, the median AF curves indicate that unstripped aircraft soot particles are not effective INPs since $RH_{hom}$ is required for ice nucleation. In contrast to unCS-soot, median AF for CS-soot exhibits a small fraction ($\leq$ 0.1 %) of the particles nucleating ice below $RH_{hom}$ at all measurements conditions except for 200 nm at 223 K (Fig. 2). The variability in the ice nucleation activity of the 400 nm particles at 218 K arises from the variability in their physicochemical properties, as will be discussed in Sect. 3.2 and 3.3. The maximum AF is also highly variable across

all measurements, ranging from 10 to 100 %. This is larger than can be explained by the uncertainty in AF and likely also reflects the variable properties of emitted particles from the different engines and thrust regimes. In the following, the ice nucleation measurements are segregated by thrust of emission and engine type in order to further understand the ice nucleation measurements shown in Fig. 2.

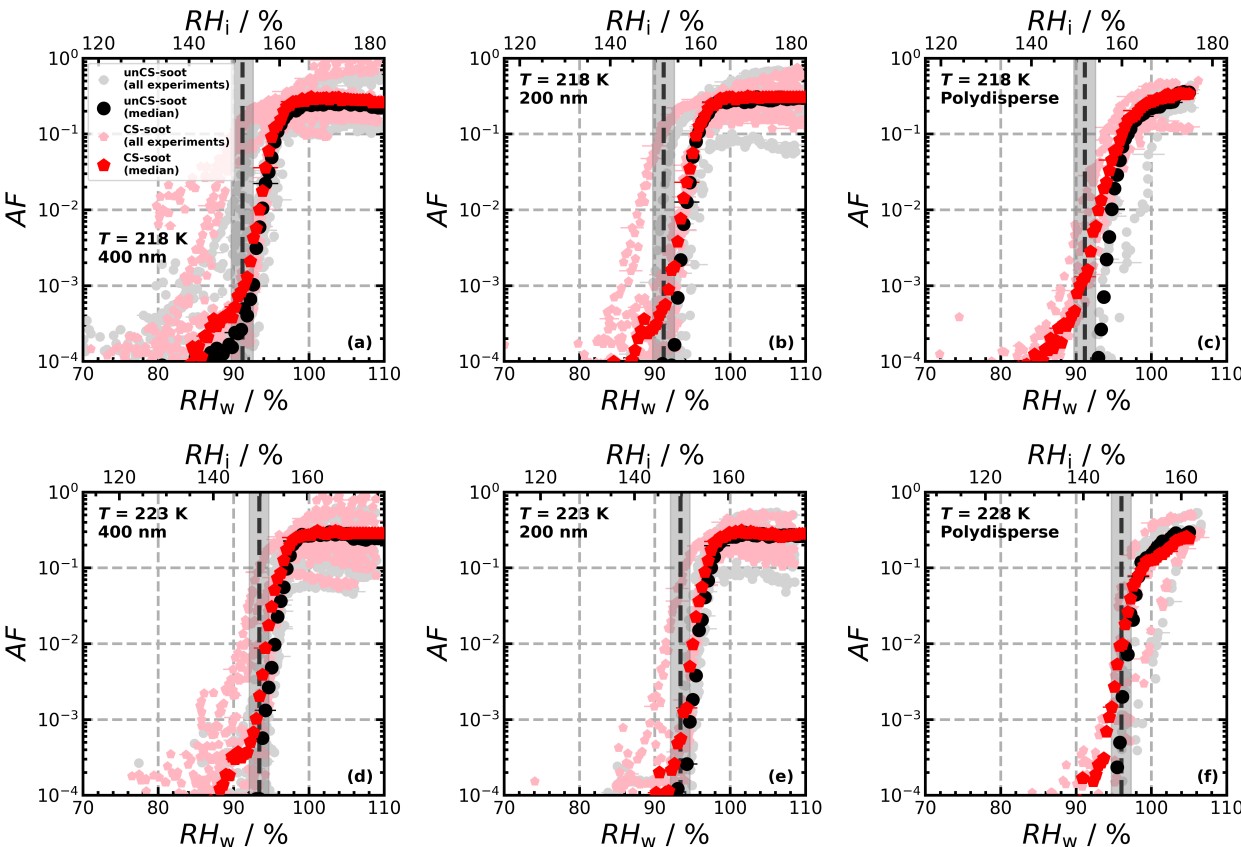

**Figure 2.** Activated fraction (AF) as a function of RH measured for the temperatures and sizes indicated in each panel. The individual experiments as well as the median AF are shown for all engine experiments. The dashed black lines indicate $RH_{hom}$, i.e., the RH at which aqueous solution droplets should freeze homogeneously (Koop et al., 2000), and the grey shaded areas represent corresponding uncertainties in HINC for our experiment conditions. Uncertainties in RH are shown only every $10^{th}$ point for clarity.





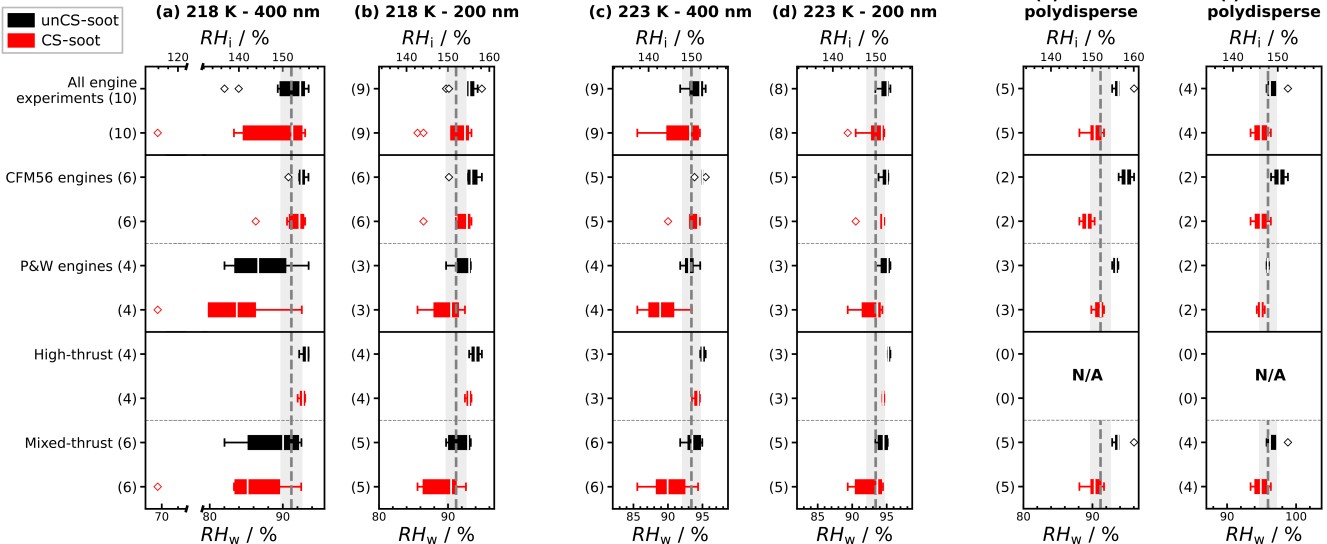

**Figure 3.** Aviation soot RH activation onset (set at AF = 0.1 %) at given temperatures and sizes. Activation onset statistics are represented with boxplots for all engine experiments and discriminated by engine thrust and type. Engines for the polydisperse experiments ran at mixed-thrust only, hence no data at high-thrust are presented. The box extends to the first and third quartile of the data, the white bar shows the median, the whiskers include 99 % of the data and the empty scatter points show the outlier data points. The dashed grey lines show $RH_{hom}$, and the grey shaded areas represent uncertainties in RH in HINC.

## 3.2 Ice nucleation ability of unCS aviation soot in relation to their morphology and mixing state

Figure 3 shows box plots summarizing the ice nucleation onset conditions of all soot samples tested, separated by thrust settings and engine type. At 218 K, unCS 400 nm aggregates ("All engine experiments" in Fig. 3a) nucleate ice around $RH_{hom}$ except for two experiments that have an onset below $RH_{hom}$. The 200 nm and polydisperse unCS particles form ice at RH > $RH_{hom}$ at 218 K (Fig. 3b). This is in good agreement with past studies showing that soot aggregate size critically affects their ice nucleation abilities (e.g. Koehler et al., 2009; Friedman et al., 2011; Nichman et al., 2019; Zhang et al., 2020a; Gao 250 et al., 2022a; Mahrt et al., 2018). As the number of primary particles generally scales with aggregate size (Sorensen, 2011), the probability for an aggregate to possess PCF relevant mesopores increases with size (Marcolli et al., 2021). The 200 nm and polydisperse soot likely have a smaller pore volume, lowering the homogeneous ice nucleation rate and the probability for pore water freezing (Marcolli et al., 2021). This likely explains why unCS 400 nm soot show modest ice nucleation in a few experiments. At 223 K and 228 K, ice nucleation onset for all sizes fall within the uncertainty range of $RH_{hom}$. With increasing 255 temperature, the size of the critical ice germ increases, and the number of pores that can host a critical ice germ decreases. The homogeneous ice nucleation rate decreases with temperature (Murray et al., 2010; Ickes et al., 2015; Marcolli, 2020), leading to a lower probability for a given soot aggregate to nucleate ice via PCF during its residence time in HINC (10 s) at e.g., 223 K compared to 218 K.



In the monodisperse experiments, both CFM56 and P&W engines (Fig.3) emit soot particles with poor ice nucleation ability,
i.e. nucleate ice homogeneously (onset AF $\approx$ RH$_{\text{hom}}$), except for P&W 400 nm at 218 K. However because 3 of the 4 P&W engines were run at mixed-thrust conditions (Fig. E2a, b and j) and only one at high-thrust (Fig. E2f), one cannot conclude a difference on engine type, because of the contribution from the thrust effect (discussed below). Experiments with the polydisperse soot particles are all mixed-thrust and overall show onset close to RH$_{\text{hom}}$, with no clear difference in AF onset between the CFM56 and P&W engines. We note that only 2 engine models were tested per engine type.

Figure 3 further reveals a clear trend of aviation soot ice nucleation ability on the engine thrust. High-thrust unCS-soot only nucleates ice at or above RH$_{\text{hom}}$ for all sizes and temperatures. Mixed-thrust unCS-soot generally also nucleates ice homogeneously but presents a larger spread of onset activation, with a couple experiments promoting ice nucleation at RH below RH$_{\text{hom}}$ for 400 nm aggregates at 218 K. In the following, the properties of the high- and mixed-thrust soot samples are investigated in view of our DVS measurements to explain their ice nucleation abilities.

The DVS isotherms of the soot samples collected onto filters are shown in Fig. 4. The mass change in percentage is defined by $\Delta m = m_0 - m_i$, with $m_0$ being the sample mass after outgassing at RH$_w$ = 0 % and $m_i$ being the mass recorded as the RH$_w$ was varied. The water uptake at RH$_w$ < 50 % indicates the sample's hydrophilicity and micro-porosity, while the isotherm behavior at higher RH$_w$ reveals characteristics of the sample meso-porosity (Haul, 1982). Both the high-and mixed-thrust soot sample are fairly hydrophobic, with less than 1 wt % adsorption at 20 % RH$_w$ (Fig. 4a and b). This indicates a paucity of
hydrophilic sites on the particle surface that could also arise from the presence of non-polar materials often associated with soot particles such as aliphatic and aromatic hydrocarbons (Popovicheva et al., 2009). However, mixed-thrust soot shows slightly stronger uptake with 4.5 % mass increase at 50 % RH$_w$ against 3 % increase for the high-thrust sample. This agrees with the observations of thrust-dependent soot properties suggesting that soot emitted at lower thrust are usually less graphitized, more oxidized (see Sect. 1) and expected to be hydrophillic (Haul, 1982) compared to soot emitted at higher thrust. Thus mixed-
thrust soot would have a higher propensity for pore filling and promoting PCF further explaining the difference observed in Fig. 3. As RH$_w$ increases, the adsorbed water molecules become active sites for further water uptake until full coverage of the particle surface (Popovicheva and Starik, 2007). Around 60 % RH$_w$, the sudden water uptake on the mixed-thrust sample indicates the merging of isolated water molecule clusters (Popovicheva et al., 2003). The strong water uptake above 80 % RH$_w$ is attributed to capillary condensation of water into the voids (pores) formed between soot primary particles and
between aggregates, followed by additional multilayer water coverage of the soot surface (Popovicheva and Starik, 2007). The presence of a hysteresis between the sorption and desorption branch for the mixed-thrust sample (Fig. 4a and c) is indicative of the presence of mesopores. Such hysteresis is very weak for the high-thrust sample (Fig. 4a and c) and indicates a Type III isotherm representing hydrophobic nonporous or macroporous particles (Haul, 1982) corroborating with their inability to promote PCF. We also note the mass loss after desorption of the high-thrust sample (-1.5 wt % at RH$_w$ = 0 %). A similar mass
loss has been observed for different soot types and explained by replacement of hydrophobic organic with water molecules (Persiantseva et al., 2004; Gubkina et al., 2001) or H$_2$SO$_4$ reactions with organic carbon condensed on the soot surface (Gao and Kanji, 2022) during the water uptake process. On the other hand, the mass of the mixed-thrust soot sample after water loss is higher than the initial mass ($\Delta m$ = 3.5 % at RH$_w$ = 0 %). This could be explained by water retained in (micro) pores due





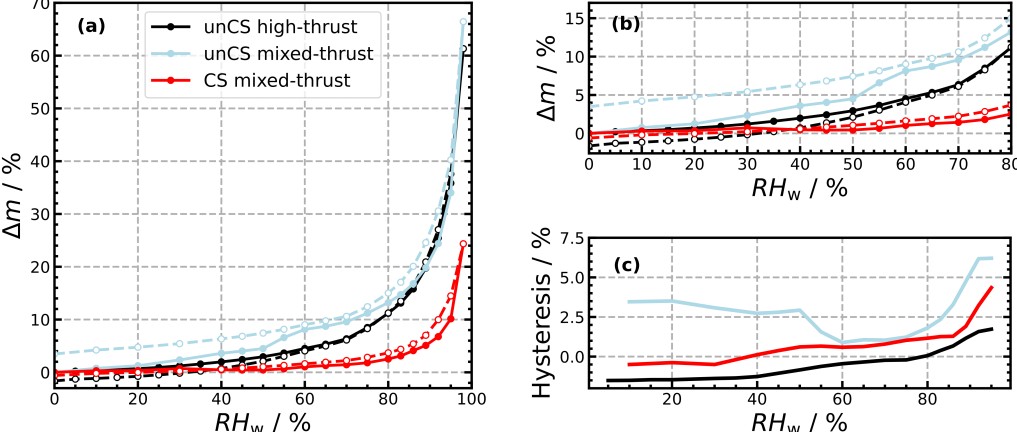

**Figure 4.** (a) Dynamic vapor sorption (DVS) isotherms of unCS high-thrust, unCS mixed-thrust and CS mixed-thrust soot samples. (b) Isotherms zoomed in the low $RH_w$ region. Open dots/dashed lines indicate desorption branch. (c) Isotherms hysteresis (i.e., desorption - sorption branch). Uncertainties are smaller than the data points, thus not visible.

to their slow desorption process, as observed by Gao and Kanji (2022) for propane soot. Overall, the mixed-thrust soot sample
could indicate a Type V isotherm with H3 hysteresis, which is common for hydrophobic macro-mesoporous adsorbents (Haul, 1982; Thommes et al., 2015).

The soot porosity can be affected by its morphology (Marcolli et al., 2021), namely primary particle diameter ($D_{pp}$) and the extent of their overlap (Brasil et al., 1999). Marhaba et al. (2019) and Liati et al. (2014) reported that average aviation soot $D_{pp}$ increases with thrust. Following this, high-thrust soot aggregates should have larger pore volume than mixed-thrust soot. This
contradicts our measurements that show a higher ice nucleation onset RH for mixed-thrust soot. In our TEM images a large range in $D_{pp}$ ($\sim$15 to $\sim$60 nm) has been observed as well as overlap between primary particles at a single aggregate level (Fig. C1) for both high- and mixed-thrust soot, limiting the $D_{pp}$ dependence on thrust level, similar to results from Vander Wal et al. (2022) and subsequent ice nucleation. Liati et al. (2014) have previously reported soot aggregates to become more compacted with increasing thrust. If such compaction causes an increase in the primary particle overlap with increasing thrust, pores might
be too narrow or closed and would support the trend in ice nucleation observed here.

In summary, the engine thrust affects the ice nucleation abilities of unCS-soot particles by controlling PCF-relevant properties. The high-thrust soot samples investigated here do not possess mesopores accessible for water condensation preventing PCF. We attribute this to poor particle wettability and strong overlap of the primary particles of the high-thrust samples (Fig. C1 panels b-e) decreasing their ice nucleation ability. We note that overlap inferred here are from 2D-images, which would
require further investigation to quantify. By contrast, the mixed-thrust soot particles are less hydrophobic and possess some mesopores likely allowing for weak PCF as observed for few experiments (Fig. 3).





### 3.3 Ice nucleation ability of CS aviation soot in relation to their morphology and mixing state

CS-soot show more variability in the ice nucleation onset compared to unCS-soot ("All engine experiments" in Fig. 3a) at all experimental conditions, however, their medians remain at or above $RH_{hom}$. Yet, considering the effect of stripping on the
monodisperse particles for each engine separately (Figs. E2-E5), it appears that stripping enhances the ice nucleation ability for a few engines. Specifically, stripping lowers the ice nucleation onset by 4 % $RH_w$ for the PW4062-3, PW4056-1C and CFM56-5B4/3 engines and by 10 % for the PW4170T engine. For the polydisperse samples, similar ice nucleation enhancement due to stripping is observed for three engines at 218 K only (Fig. E1a, c and d). The effect of stripping different sized populations on ice nucleation does not follow the effect of size on ice nucleation, i.e., stripping had no effect for the PW4062A-3 soot
which had the largest size mode (289 nm), while the ice nucleation properties of the smaller PW4168A #1 soot (121 nm) were sensitive to stripping and show moderate ice nucleation below $RH_{hom}$ at 218 K, underlying the importance of particle mixing state.

Discriminating by engine type, the CS-soot shows similar results as discriminating by thrust. As for the unCS-soot, due to the thrust confounding factor (3 out of the 4 P&W engines run at mixed-thrust), the engine type effect on the ice nucleation
of CS-soot is not elucidated. For high-thrust conditions, CS-soot initiates ice nucleation at or above $RH_{hom}$ for all sizes and temperatures (Fig. 3). For mixed-thrust conditions CS-soot generally nucleates ice at lower RH with a median onset RH slightly below $RH_{hom}$ for 400 nm particles. For the polydisperse particles, the smaller onset spread compared to size-selected particles results from the smaller fraction of large particles (Appendix D).

DVS isotherms of the CS mixed-thrust sample is shown on Fig. 4a and b. Overall, the water uptake is weak, below 1
% mass at 50 % $RH_w$, compared to 4.5 % for the unCS mixed-thrust samples. This weak water uptake is indicative of a very hydrophobic surface where the water molecules mostly interact with the surface through low energy dispersion forces (Popovicheva et al., 2011). TGA performed on the mixed-thrust sample, mimicking catalytic stripping, led to a mass loss of 10 % upon heating at 623 K (Fig. G1). Loss of sulfates and volatile material might explain the low water uptake capacity after heating. In particular, from 298 K to 373 K, the mass loss is about 1.5 %, and can be attributed to water and $H_2SO_4$ desorbing
from the surface and pores. The remaining mass loss occurring at higher temperature indicates desorption of high- to moderate-volatile materials such as low-molecular-mass PAHs (Portet-Koltalo and Machour, 2013) and larger hydrocarbons. Removal of $H_2SO_4$ would decrease particle hygroscopicity (Popovicheva et al., 2011). While removal of PAHs and other organics usually render soot particles more hydrophilic (Persiantseva et al., 2004; Demirdjian et al., 2009), aliphatics and PAHs found on aviation soot particles emitted at mixed-thrust conditions (Abegglen et al., 2016; Marhaba et al., 2019) can be oxidized when
stripped at 623 K in air (Stanmore et al., 2001), increasing the number of surface sites for water uptake. Given the opposite effects on water uptake, we attribute the decreased water uptake capacity of the CS mixed-thrust sample (Fig. 4) to sulphate removal. Capillary condensation for the CS mixed-thrust sample is indicated by the steep water uptake above 80 % $RH_w$ (Fig. 4c), similar to the unCS mixed-thrust sample. However, the hysteresis magnitude is reduced for the CS-soot sample, indicating a smaller pore volume is available for water condensation. Results from our TEM-based soot morphology analysis presented in
Fig. F1, indicate no particle restructuring after stripping, consistent with the results of Bhandari et al. (2017). This indicates that





the morphology cannot explain the change in hysteresis upon stripping, nor the enhanced ice nucleation ability observed for CS-soot. Furthermore, a more hydrophobic surface, associated with a removal of sulfur, can inhibit the filling of mesopore by water vapor until sufficiently high $RH_w$ values are reached, supporting poor ice nucleation ability through PCF (Marcolli et al., 2021), which would be opposite to the trend observed in the ice nucleation measurements which show enhanced ice nucleation

onset for CS-soot. This can be reconciled by considering that removal of $H_2SO_4$ increases the homogeneous nucleation rate of the pore water, and thus increases the number of mesopores that can freeze at the investigated temperatures during the aerosol residence time. This is further discussed in Sect. 3.4. Due to a lack of sample mass, DVS analysis could not be conducted for CS high-thrust soot. Nonetheless, the DVS analysis revealed a mesopore paucity for the unCS high-thrust soot sample (Sect. 3.2) which exhibited poor ice nucleation ability similar that of high thrust CS-soot. This suggests that sulfur stripping did not

enhance the ice nucleation ability of high thrust samples (as shown in Fig. 3) also because of mesopore paucity. Moreover, high-thrust soot particles are organic carbon poor (Abegglen et al., 2016), and therefore would be insensitive to heat treatment by catalytic stripping (Ess et al., 2016) supporting a negligible effect of stripping.

In summary, the different properties of the unCS and CS high- and mixed-thrust soot samples highlight the dominant role of mesopore availability for PCF. Similar to previous studies (Koehler et al., 2009; Lupi et al., 2014; Mahrt et al., 2020a), we

emphasize that particle water uptake capacity alone is not a sufficient predictor for the ice nucleation ability of soot particles. To complement to the bulk analysis of the samples, the chemical properties of the mixed-thrust aviation are explored in the following on a single particle level.

## 3.4    Ice nucleation measurements of selected engines emitted at mixed-thrust in relation to their chemical and physical properties

To better understand the distinct ice nucleation ability at $RH < RH_{hom}$ of mixed-thrust aviation soot (Fig. 3), we analysed the chemical composition of soot from 6 engine types using STXM/NEXAFS and STEM-EDX (Fig. 5) whose ice nucleation AF curves are shown of Fig. 6. Generally, common features can be identified in the spectra of all engines, such as the strong peak in the carbon K-edge (C-edge) spectra at 285.4 eV due to $\pi^*$ excitation of aromatic C=C in graphite, typical for soot particles (Hopkins et al., 2007; Moffet et al., 2011). X-ray absorption peaks between 286-290 eV are indicative of absorption

from organic carbon (Moffet et al., 2011). At energies 288-289 eV, $\sigma^*$ absorption by C-H groups occurs (Bruley et al., 1990). Oxygen NEXAFS were also measured (Fig. H1) and show typical peaks for C=O functionalities around 531 eV and C-O and C=O peaks around 538 eV (Moffet et al., 2011). Sulfur should also contribute to the oxygen spectra at 538 eV (Zelenay et al., 2011), but only for unCS-soot particles as CS-samples should be depleted of sulfur, as observed in the STEM-EDX measurements (Fig. 5c). Overall, a higher sulfur content should coincide with higher oxygen content in unCS-soot (Fig. 5c).

However, we do find some stark differences in STXM/NEXAFS and STEM-EDX results that could explain the ice nucleation ability of soot emitted from different engines. These are discussed in the next 3 sections in 3 different cases indicated in Figs. 6 and 5. Additionally, we computed the ratio between absorption at 285.4 eV and 292 eV, $R$, as an indicator of the degree of graphitization of the sample, e.g. where the values $R = 1.55$ and $0.62$ correspond to graphite and soot respectively from a CFM56-7B26 engine previously investigated in Liati et al. (2019). We note that stripping should remove surface bound volatile



compounds and the degree of bulk graphitization should not change. Therefore, as STXM/NEXAFS is a bulk sensitive method, we do not expect $R$ to vary significantly (comparable within uncertainty) between unCS- and CS-soot particles as shown in Fig. 5. However, there are differences in $R$ for soot emitted from different engines which are discussed below.

### 3.4.1 Case 1: Oxidized aviation soot particles are weak INPs

Figure 6a-h show the ice nucleation AF curves for the PW4056-1C and CFM56-5B4/3 engines. 1 % (0.1 %) of the unCS 400
nm soot particles from the PW4056-1C engine nucleate ice at RH < $RH_{hom}$ at 218 K (223 K) (Fig. 6a and b). The concave-like increase in AF below $RH_{hom}$ is attributed to pores of various sizes/contact angles triggering PCF at different RH thresholds (David et al., 2020). By contrast, unCS 400 nm soot particles from the CFM56-5B4/3 engine and 200 nm unCS particles from both engines did not exhibit PCF. Soot samples from both engine types show a similar response to stripping: enhanced ice formation with the ice nucleation onset below $RH_{hom}$ at all measurement conditions. An exception are 400 nm PW4056-1C
soot at 218 K, which already nucleated ice at $RH_{hom}$ for the unCS experiment.

STXM/NEXAFS spectra of soot particles for Case 1 are shown in Fig. 5a[1]. A peak at 287.5 eV is visible for both engines and attributed to aliphatic-C (Hopkins et al., 2007; Parent et al., 2016), due to the fact that the oxygen spectra (Fig. H1a) have a broad peak around 538 eV while lacking a major carbonyl transition at 531 eV (Hopkins et al., 2007; Parent et al., 2016). This is observed only to a minor extent for the PW4056-1C engine. We speculate this type of organic matter present may fill
pores preventing PCF and could explain why ice nucleation below $RH_{hom}$ is hardly visible for unCS-soot (Fig. 6a-h). A minor peak at 288.6 eV in the PW4056-1C spectra indicates the presence of (hydrophilic) carboxyl groups (Hopkins et al., 2007; Parent et al., 2016). In this case, $R$ (Fig. 5b) is lower compared to the CFM56-7B26, PW4062A-3 and PW4168A #2 engines (engines discussed in Sect. 3.4.3 below). Lower graphitization, in turn, can lead to a higher reactivity toward oxygen (Liati et al., 2019). CS-soot show similar absorption peak locations, although slightly reduced compared to absorption at 285.4 eV.
This can result from the oxidation and evaporation of aliphatics (Raj et al., 2013). For both engines, oxygen and sulfur content revealed by STEM-EDX (Fig. 5c) is highest amongst the engines considered in the 3 cases presented here. Figure 5c also shows that sulfur is largely removed on CS-soot. Sulfur promotes water condensation in pores by lowering the contact angle but it also decreases the pore water activity and the homogeneous freezing rate coefficient $J_{hom}$ (Koop et al., 2000), ultimately hindering PCF at lower RH (Gao et al., 2022c). In particular, the presence of $H_2SO_4$ can decrease $J_{hom}$ by several orders of
magnitude, as illustrated in Fig. J1. The sulfur content of the PW4056-1C engine is about 0.11 atom. % ($\approx$ 0.28 wt %), which is higher than that of the coated propane soot (0.10 wt %) studied in Gao et al. (2022c), with the same technique. The authors reported the poor ice nucleation ability of these particles owing to the presence of $H_2SO_4$. The higher sulfur content in our samples explains the poor ice nucleation ability for unCS-soot and enhancement for CS-soot (Fig. 6a-h and Sect. J.) This is

---

[1]We note that data for a CFM56-5B3/3 instead of a CFM56-5B4/3 engine are shown in Figure 5, because a STXM/STEM sample could not be collected for the latter. CFM56-5B4/3 and CFM56-5B3/3 engines are the same engine model and both have a Tech Insertion combustor for lower $NO_x$ emission. The differences are the engine age and thrust rating (thrust delivered at each engine power regime), with slightly higher thrust rating for the CFM56-5B3/3 engine (ICAO, 2023). While this results in higher soot emission for the CFM56-5B3/3 engine, especially at high thrust, we expect the soot particles of both engines to have similar physicochemical properties.





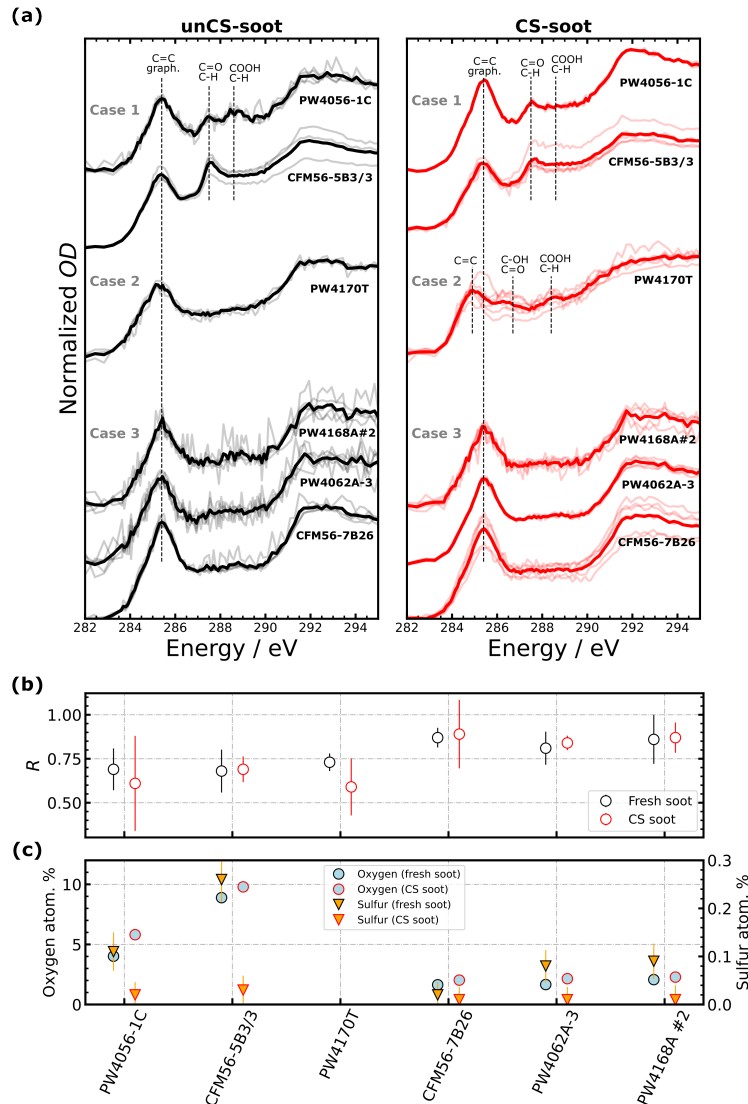

**Figure 5.** (a) Near edge x-ray absorption fine structure (NEXAFS) spectra for the given engine types. Bright and pale lines correspond to mean and single particle spectra respectively. The spectra of the different engine types are offset vertically for clarity (b) 285.4 eV/292 eV carbon K-edge peak height ratio $R$ indicating the graphitization level of the sample. Error bars are 1 $\sigma$. (c) Oxygen and sulfur soot atomic concentration (in percentage of total element concentration, atom. %) for indicated engines derived from elemental mapping of soot particles with STEM-EDX (see exemplary images in Fig. I1). Due to an insufficiently loaded TEM grid only STXM and no STEM-EDX analysis could be performed for the PW4170T engine sample (missing data points). Error bars show 1 $\sigma$. Uncertainties in oxygen content are not visible as they are smaller than the data points. Low nitrogen content (~1 atom.%, as well as trace of Sn, P and Al) were observed similarly on all samples, thus contribute systematically to the particle properties and are not shown.





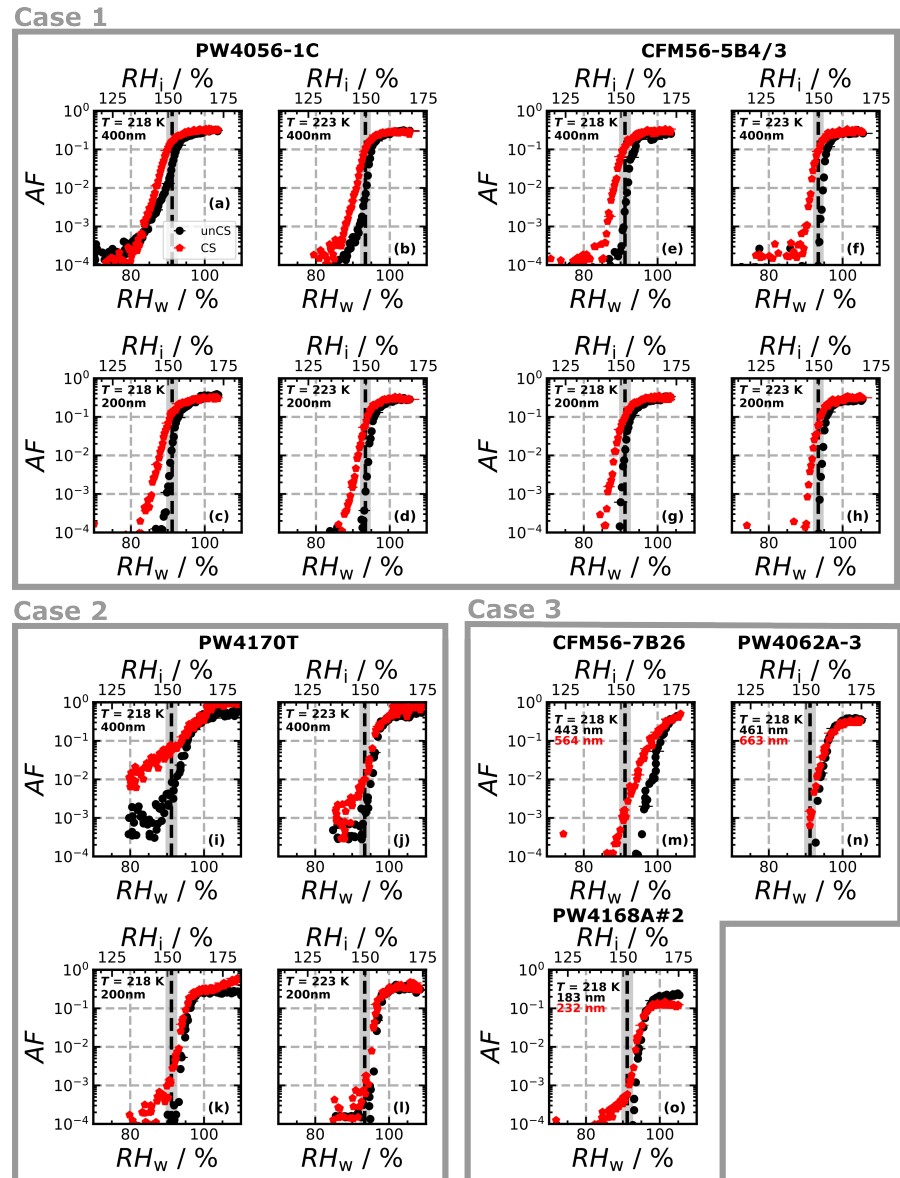

**Figure 6.** Ice nucleating activated fraction (AF) for the given engine types at the given temperatures and sizes as a function of RH. For Case 3, the minimum diameter of the 0.1 % largest particles is indicated in the legend. Corresponding NEXAFS spectra, $R$ ratio and element composition are shown in Fig. 5. (a) shares its legend with the other panels.

.

also corroborated by the observation that temperature had little to no effect on the ice nucleation onset RH of the CS-soot, with 410 0.1 % AF reached at 140 % $RH_i$ at both 218 and 223 K for 400 nm PW4056-1C soot and around 145 % $RH_i$ for CFM56-5B4/3 soot. This is an indication that PCF is not limited by ice nucleation of aqueous $H_2SO_4$/sulphate solution in the pores.



The above engines were also investigated for aggregate morphology. Our TEM image analysis revealed that CFM56-5B3/3 aggregates are on average more compact and spherical compared to PW4056-1C soot (Fig. F1). At first sight this is in contrast to the ice nucleation results which are enhanced for the less compact unCS 400 nm PW4056-1C soot particles (Nichman et al., 2019; Mahrt et al., 2020b; Gao and Kanji, 2022). Closed compact structure in aggregates increases the probability of cavities with the right properties for PCF. However, when considering the overlap of the primary particles as well as their diameter, the overall pore (cavity) volume can become lower for compact aggregates (Marcolli et al., 2021). The TEM images for both engines reveal highly overlapping primary particles (Fig. C1f-o) but generally more fused and almost fully overlapping for the CFM56-5B3/3 engines which leads to pores of smaller sizes compared to the PW4056-1C engine, contributing to its poorer ice nucleation abilities. We note that the spread of primary particle diameters observed in Fig. C1 likely translates into a considerable variety of pore structures, with different sizes causing ice formation via PCF at different RH thresholds. This is consistent with the concave-like shape of the AF curve for 400 nm PW4056-1C soot at 218 K, where a steep, step-like increase in AF is absent that would indicate PCF by similar pore properties across the soot aggregates.

Overall the chemical analysis reveals the presence of surface defects and hydrophilic functional groups for the PW4056-1C and CFM56-5B4/3 engines, likely facilitating PCF. Differences in the the ice nucleation between engines come from morphological and mixing state properties. The removal of sulfur explains the enhanced ice nucleation of the CS samples.

### 3.4.2 Case 2: Low NO$_x$ combustor soot particles (PW4170T engine) are moderate INPs

Figure 6i-l show the ice nucleation AF curves for the PW4170T engine type. Due to a low number concentration of emitted particles, coagulation in the aerosol tank was weaker and the concentration of 400 nm particles remained low. For this reason, the instrument detection limit is about AF = $10^{-4}$-$10^{-3}$ at this size ($< 10^{-4}$ for all other ice nucleation measurements). unCS-soot from the PW4170T engine shows a similar ice nucleation activity compared to PW4056-1C, with weak PCF for 400 nm at 218 K, and homogeneous freezing for the other measurement conditions. However, CS-soot from the PW4170T engine showed a significant enhancement of the AF below RH$_{hom}$ for 400 nm at 218 K (Fig. 6i). In comparison to other engines, the PW4170T engine is equipped with a TALON IIB combustor designed for low NO$_x$ emission (McKinney et al., 2007), but also showed the lowest soot emission index as measured during the engine experiments.

NEXAFS spectra for the unCS-soot (Fig. 5a) show no strong peaks over the 286-290 eV region as opposed to Case 1. However, absorption progressively increased in this energy range and may indicate multiple organic functionalities. The presence of oxygen for the unCS-soot was confirmed at the O-edge (Fig. H1a) with a clear absorption around 538 eV (Fig. H1b). Upon stripping, the $\pi^*$ graphite peak is reduced and shifted to lower energy at 284.9 eV although we note that one spectra (i.e., 1 particle) showed a graphite peak at 285.4 eV similar to the unCS-soot. Parent et al. (2016) attributed the transition at 284.9 eV to excitation of $\pi^*$ C=C bonds other than graphite, e.g. of surface organics such as PAHs which should have been desorbed upon stripping, decreasing their contribution to absorption. Charring (pyrolysis) of non-refractory organics is thought to be unlikely at the investigated temperatures and thus does not explain neither the change in C=C absorption (Ess et al., 2016; Bhandari et al., 2017). Jäger et al. (1999) measured similar spectral shape for graphite electrode soot and attributed the reduced $\pi^*$ graphite transition and less prominent $\sigma^*$ graphite band at around 292 eV to overall weakly graphitized soot. Such change





in carbon bonds upon heating around 623 K has been observed for organic-rich diffusion flame soot as the result of removal of surface aromatics in condensed volatile organic material and the formation of more saturated bonds (Ess et al., 2016; Gao et al., 2022b). An increase in excitation of oxygenated functional groups is also observed, namely two peaks at 286.7 eV and 288.4 eV corresponding to phenolic-C/ketone and carboxyl, respectively (Hopkins et al., 2007; Parent et al., 2016). The presence of

ketone after stripping is suggested at the O-edge (Fig. H1b), however only one spectrum was obtained for this sample.

The significant change in surface properties of PW4170T soot after stripping is supported by the large mass fraction stripped out of 200 nm particles, i.e., ∼10 %, compared to ∼3 % for other mixed-thrust engine tests (measured by the CPMA, Table K1). Due to the lower emission index of the PW4170T engine compared to others, we hypothesize that a reduced overall soot surface area onto which semi-volatile compounds could condense, leads to an increase in soot coating compared to other

engines. However, a detailed particle and gas phase characterization at the engine exit plane beyond the scope of this study would be needed for verification.

The fractal dimension of soot aggregates from the PW4170T engine is slightly larger compared to other engines for which mass measurements were performed (Table K1) suggesting that PW4170T soot were overall more compact. As mentioned earlier, compact aggregates, in particular for CS-soot, can have larger mesopore volume compared to open-branched soot, thus

can be potent INPs via PCF (Gao et al., 2022a). The response of PW4170T engine soot to stripping can be explained by a high reactivity toward oxygen due to low crystallinity. Oxidation suggests the particles are more hydrophilic further supporting their enhanced ice nucleation abilities.

### 3.4.3 Case 3: Highly graphitic and poorly oxidized soot particles are ineffective INPs

The CFM56-7B26, PW4062A-3 and PW4168A #2 (hereafter PW4168A) engine unCS-soot particles all nucleate ice at $RH_{hom}$

or higher (Fig. 6m-o). Yet their response to stripping varies, with a small enhancement at low AF for two out of the three engines. This could be due to the maximum particle size in the CS experiments being higher than in the unCS experiments or due to an increase in size with time from coagulation in our mixing tank, given the CS was conducted after the unCS-soot experiments. NEXAFS spectra (Fig. 5a) showed no strong organic absorption peaks over the 286-290 eV energy range and no absorption at the O-edge (Fig. H1c). Values of $R$ are slightly higher compared to the other engine types discussed above,

i.e. about 0.89, which indicates fairly graphitized particles. The weakly oxidized nature of the unCS particles is confirmed by STEM-EDX analysis. The oxygen content of the unCS and CS samples are ≤ 2.5 atom. % (Fig. 5c). The sulfur content of the unCS-soot is about 0.08 atom. % for two engines and 0.02 atom. % for the CFM56-7B26, that might explain the higher RH onset for the unCS particles of the latter (Fig. 6m). Similarly as for the PW4056-1C and CFM56-5B3/3 engines, sulfur is largely removed upon stripping (< 0.01 atom. %). The poor ice nucleation ability of the unCS-soot and the limited effect of stripping

on the particles can likely be explained by the initial high graphitic nature of the particles, resulting in low reactivity toward oxygen and poor surface water uptake capacity. The small ice nucleation enhancement for the CFM56-7B26 and PW4168A engine could be due to removal of organic material and sulfur, making pores available for PCF, as observed for thermo-denuded organic-rich propane soot in Gao et al. (2022b).



## 4   Atmospheric implications

Previous studies have often used proxies of aviation soot particles (e.g. Zhang et al., 2020a; Gao et al., 2022b; Nichman et al., 2019; Mahrt et al., 2018) to infer the ice nucleation ability of aviation soot. Emissions from real jet engines could have vastly different properties as demonstrated in this work.

Overall, we find real aviation soot particles to be ineffective INPs at cirrus temperatures. Our experiments revealed that most aviation soot particles tested here only formed ice crystals very close to or above RH conditions required for homogeneous

freezing of aqueous solution droplets where competition of freezing from typical background upper tropospheric aerosol is high (Schröder et al., 2002). We therefore expect aviation soot particles to have a negligible effect in modifying cirrus cloud properties and formation pathways via their role as INPs.

For example, mineral dust particles typically require considerably lower humidities (e.g., about 120 % RH$_i$ at 220 K; Ullrich et al., 2017) to promote heterogeneous ice nucleation compared to aviation soot particles tested here. Given that mineral dust

particles are present throughout the upper troposphere (Froyd et al., 2022), and despite their generally much lower number concentration compared to that of aviation soot close to aircraft flight corridors, we expect ice nucleation on mineral dust to be dominant. This has been demonstrated in a recent modelling study where for a large range of conditions and combinations (updraft velocities, dust and soot particle number concentration), ice nucleation on mineral dust almost always out-competes and even in some cases suppresses that from aviation soot particles due to consuming available humidity (Kärcher et al., 2023).

Soot particles emitted by aviation have diameters in the range of 30 to 40 nm at cruise thrust (Moore et al., 2017; Durdina et al., 2021b). Ice nucleation is strongly size dependent (Mahrt et al., 2018), and given that 200 nm size selected particles in this work were poor INPs, we further strengthen the claim that aviation soot will not perturb background cirrus cloud formation, except via contrail formation. Furthermore, we investigated polydisperse soot populations with mode diameters from 70 to 450 nm, which also exhibited ice formation at or above RH conditions relevant for homogeneous freezing of aqueous solution

droplets (Fig. 3e, f). Only a handful of older studies reported soot particles larger than 200 nm in real-world aviation plumes (Petzold et al., 1999; Ström and Ohlsson, 1998). Newer more recent studies show typical aircraft emitted soot particles to be in the size range below 50 nm (Moore et al., 2017; Durdina et al., 2021b). We sampled 400 nm particles which were achieved by coagulation in an aerosol tank prior to ice nucleation measurements to further elucidate the particle size effect. A striking outcome is that even the 400 nm particles were on average ineffective INPs except in isolated cases showing moderate ice

nucleation ability (mixed-thrust and CS-soot). Previous studies with propane soot and commercially available carbon blacks (Gao et al., 2022a; Mahrt et al., 2018; Nichman et al., 2019; Koehler et al., 2009) have shown large sizes to be effective or even potent INPs. This illustrates that the physical and chemical particle properties of aviation soot cannot be represented by proxies. One difference for example, is that aviation soot primary particles are generally fused and highly overlapped, limiting the appearance of cavities responsible for triggering ice nucleation via PCF. The high graphitic nature of aviation soot further

constrains their ice nucleation ability by limiting their water uptake (Marcolli et al., 2021).

The mixing state impacts on ice nucleation were studied here via catalytic stripping to denude organics and sulfur (condensables), and STXM/NEXAFS and STEM-EDX to characterise their removal. CS-soot in general revealed an enhancement in





ice nucleation. We interpreted this as removal of condensables blocking the pores on the unCS-soot aggregates limiting their availability for PCF. In the sampling system used here, the formation of nucleation mode particles from sulfur and organics

was curbed by keeping the exhaust at 433 K and drying it before feeding the aerosol tank. In the atmosphere, we expect real world aircraft soot particles to be equivalent to, or even more coated that our unCS-soot sample, i.e. be internally mixed with even higher fractions of organics and sulfur (Kärcher et al., 2007). This implies that even the moderate ice nucleation activity observed here could be an over estimate of the aviation soot in emitted in the upper troposphere.

Finally, across all engine types, thrust levels and cirrus temperatures tested here, we found aviation soot to reach a maximum

AF of 1 % prior to the onset of homogeneous freezing of aqueous solution droplets (see Fig. 6a, unCS-soot). Thus only a small fraction of ambient aviation soot particles can act as INP at cirrus conditions, further supporting their negligible role for aerosol-cloud interactions. In fact, this observation is in very good agreement with the findings of Kärcher et al. (2021), who estimated that less than 1% of aircraft emitted soot particles can nucleate ice via PCF alongside homogeneous freezing of aqueous solution droplets.

The overarching theme of our results is that aviation soot particles from the combustion of Jet A1 fuel, are not cirrus relevant INPs. Further they categorically show that soot particles from real aircraft engines can have vastly different ice nucleation abilities compared to soot particles from other emission sources. In particular, the low ice nucleation activities found here suggest that previous estimates of radiative forcing associated with aircraft soot-cirrus interactions summarized by Lee et al. (2021) can be biased and need to be critically reassessed, as the parameterizations underlying soot ice nucleation were often

based on laboratory proxies of aircraft soot rather than particles from real aircraft engines. This also implies that with future changes in fuel type such as sustainable aviation fuel and low emission index engines, such ice nucleation assessments will be needed.

## 5 Conclusions

In this work we investigated the ice nucleation abilities and physicochemical properties of soot particles emitted from real

aircraft engines during engine testing and maintenance, in an attempt to provide a first best estimate of their role as INPs at cirrus cloud conditions. Particles sampled were emitted from CFM International and Pratt & Whitney engines. The ice nucleation measurements were conducted at cirrus temperatures of 218 K, 223 K and 228 K for polydisperse and size-selected soot particles of 200, 400 nm diameter, and as a function of engine thrust (power), using a continuous flow diffusion chamber.

Particle morphology, mixing state, surface functionalities and wettability were investigated by means of various techniques.

Aviation soot particles were overall poor INPs at cirrus temperatures: soot emitted at high-thrust were found to always nucleate ice at relative humidity (RH) $\geq$ RH$_{hom}$, i.e. conditions characterizing the homogeneous freezing of aqueous droplets. Similarly, soot emitted at mixed-thrust generally nucleated ice at RH $\geq$ RH$_{hom}$. In a few experiments, ice nucleation via PCF at RH $<$ RH$_{hom}$ for 400 nm size-selected aggregates is observed. The absence of PCF for the high-thrust aviation soot is attributed to a paucity of mesopores, likely due to a non-uniform primary particle diameter and generally highly fused (overlapping)

primary particles, both which reduce mesoporosity. Mixed-thrust soot, by contrast, possessed some mesopores and are less



hydrophobic, as revealed by DVS, likely explaining their slightly better ice nucleation abilities via PCF. Interestingly, high-thrust emitted soot ice nucleation ability is insensitive to catalytic stripping. This is explained by their highly graphitized nature and lack of mesopores. In contrast, catalytically stripped mixed-thrust soot are able to promote weak PCF for 400 nm and 200 nm soot. We attribute this to the removal of sulfur condensed into the soot pores, that would otherwise lower the homogeneous ice nucleation rate of the pore water, lowering its ability to promote PCF as in the experiments without catalytic stripping. Targeted analysis of selected soot populations by STXM/NEXAFS and STEM-EDX revealed that the modest variability in aviation soot ice nucleation can likely be explained by different surface chemistry, namely the presence of hydrophilic sites. We point out that the overall ice nucleation abilities of real aviation soot was found to be different to that of propane flame soot particles, which has previously been used as a surrogate for aviation soot (e.g. Mahrt et al., 2018; Marhaba et al., 2019; Ikhenazene et al., 2020). Such differences are likely due to the presence of sulfur and the different primary particle properties mentioned above. Overall, the results indicate that aviation soot particles tested here nucleate ice at or very close to relative humidity required for homogeneous freezing of solution droplets, the latter being ubiquitous in the upper troposphere (Schröder et al., 2002). With the more effective ice nucleation abilities of mineral dust, also present at typical flight altitudes (Froyd et al., 2022) the effect of aviation soot will be limited (Kärcher et al., 2023). To further understand the ice nucleation ability of (aviation) soot particles, quantitative measurements of soot primary particle size and overlap would be needed. Developing parameterizations that connect the effect of aqueous solution freezing in pores would allow better predictive capability of aviation soot ice nucleation.

*Data availability.* The data presented in this work can be found at DOI: 10.3929/ethz-b-000634341. The link will be activated upon final acceptation of the manuscript.



**Appendix A: Aircraft engine emission procedure**

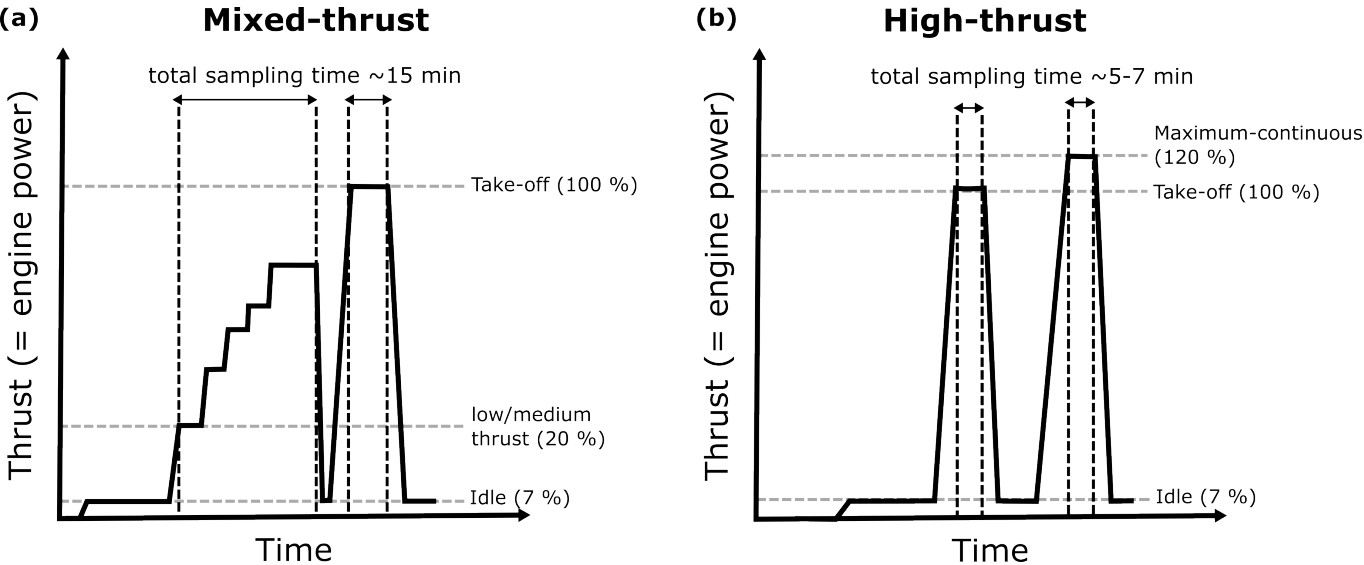

**Figure A1.** Engine power (thrust) procedure followed during the engine maintenance tests. The values of (sea level) engine thrust are rough estimates of the thrust reached. Procedure (a) is referred as "mixed-thrust" experiment while procedure (b) as "high-thrust" experiment.





**Appendix B: Measurement overview**

Table B1 gives an overview of the engines tested in this study and the various associated measurements. A total of 14 engines were tested for which ice nucleation experiments were conducted. The particle size (200 nm and 400 nm monodisperse or polydisperse) apply to the ice nucleation measurements, the size distribution measurements and for the particle sampling onto
TEM grids used for analysis by TEM, STEM-EDX and STXM/NEXAFS. As we were limited to the operating procedure of the engine test cell facility, mass measurements and grid collection could not be conducted for all tested engines.





**Table B1.** Measurements overview. Numbers in the columns "TEM images", "STEM-EDX mapping" and "STXM/NEXAFS" correspond to the numbers of images, maps and spectra collected per sample (left hand side value corresponds the unCS-soot sample and right hande side value to stripped soot sample).

| Engine model | Thrust (%) | Size (nm) | TEM images | STEM-EDX mapping | STXM/ NEXAFS | Mass data[e] |
|---|---|---|---|---|---|---|
| PW4062-3 | 30-100 | Mono.[c] | – | – | – | – |
| CFM56-5B3/3[a] | 30-100 | Mono.[c] | **100**, **120** | **9**, **7** | C: **3**, **5** O: **3**, **2** | – |
| PW4056-1C | 30-100 | Mono.[c] | **112**, **97** | **7**, **7** | C: **4**, **2** O: **2**, **2** | – |
| CFM56-5B4/3 | 30-100 | Mono.[c] | – | – | – | – |
| CFM56-5B3/P | 80-120 | Mono.[c] | – | – | – | $\Delta m$ |
| CFM56-7B24E | 80-120 | Mono.[c] | – | – | – | $\Delta m$ |
| PW4056-3 | 80-120 | Mono.[c] | – | – | – | $\Delta m$, $D_{fm}$ |
| CFM56-5B3/P | 80-120 | Mono.[c] | – | – | – | $\Delta m$, $D_{fm}$ |
| CFM56-7B24 | 30-100 | Mono.[c] | – | – | – | $\Delta m$, $D_{fm}$ |
| CFM56-7B26/3 | 30-100 | Mono.[c] | – | – | – | $\Delta m$, $D_{fm}$ |
| PW4170T | 30-100 | Mono.[c] | – | – | C: **3**, **6** O: **1**, **1** | $\Delta m$, $D_{fm}$ |
| CFM56-7B26 | 30-100 | Poly. (180-263)[d] | **132**, **131** | **9**, **9** | C: **5**, **6** O: **2**, **1** | – |
| PW4062A-3 | 30-100 | Poly. (195-289)[d] | **138**, **143** | **5**, **10** | C: **5**, **4** O: **2**, **2** | – |
| CFM56-7B26E | 30-100 | Poly. (188-246)[d] | – | – | – | – |
| PW4168A #1[b] | 30-70 | Poly. (84-121)[d] | **108**, **111** | **7**, **10** | – | – |
| PW4168A #2[b] | 30-100 | Poly. (107-133)[d] | **120**, **121** | – | C: **5**, **5** O: **2**, **1** | – |

[a] no ice nucleation data for this engine, [b] same engine tested on different days and for different engine power, [c] particle size-selected at 200 nm and 400 nm, [d] polydisperse particles and associated particle mode diameter determined with SMPS (Fig. E1 panels K-o), [e] $\Delta m$ = particle mass change after stripping and $D_{fm}$ = mass-mobility exponent (displayed in Table K1)



## Appendix C: HRTEM imaging of aviation soot particles

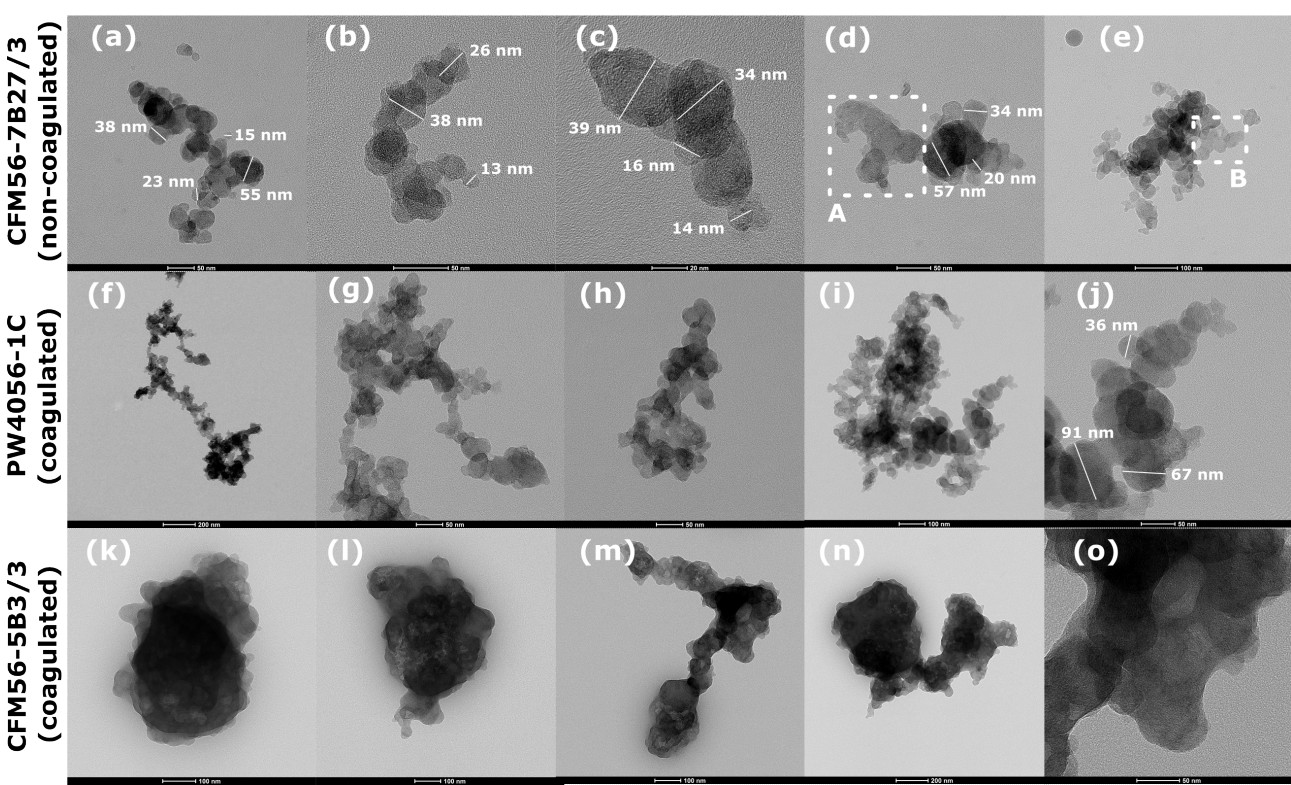

**Figure C1.** (a-e) Soot particles collected downstream of the engine where the exhaust is rapidly diluted with air, hence undergoing no coagulation process (position 1 in Fig. 1). Particles there were emitted mostly at high-thrust. (f-j) Mixed thrsut unCS coagulated 400 nm PW4056-1C soot (collected downstream of the aerosol tank, position 2 in Fig. 1). (k-o) same as (f-j) for CFM56-5B3/3 soot particles. Panels b-d and j show typical primary particle sizes within the soot aggregates. Regions with equal contrast (equal grey scale) indicate particle region with equal thickness. For instance, regions A in panel d and B and panel e indicate highly fused primary particles.





## Appendix D: Aviation soot size distributions and size effect on the ice nucleation abilties

Figure D1 shows size distributions of monodisperse size-selected 400 nm aviation soot and a typical polydisperse size distri-
bution of aviation soot coagulated after several hours in the aerosol tank. The size distribution of real insitu aviation soot sizes
measured by Moore et al. (2017), is shown for comparison. Size-selected particles with the DMA are not entirely monodisperse
but present a fraction of larger particles that correspond to multiple-charged size selected particles (Flagan, 2011). Fraction of
multiple-charged (mostly double-charged) size selected particles for a 400 nm size-distribution is about 10 % in our study with
mode size $\approx$ 660 nm (Fig. D1). Assuming that the ice-active aviation soot particles nucleate ice via PCF, we can hypothesise
that the largest particles nucleate ice at lower RH than smaller ones (they have a higher probability to possess mesopores suit-
able for PCF; Mahrt et al., 2018). We observe 0.1 % to 1 % of the 400 nm soot population nucleating ice below $RH_{hom}$ (Fig.
2a and E2), applying this percentage to the 400 nm size distribution implies that only particles with diameters > 800 nm and >
730 nm respectively would nucleate ice (Fig. D1). The fraction of particles > 730 nm and > 800 nm from the polydisperse size
distribution correspond to only 0.035 to 0.0025 % of the soot population (Fig. D1). This is below or just at the detection limit
of the instrument (about 0.01 %) explaining the relatively small ice-active fraction (if any) of the unCS polydisperse particles.

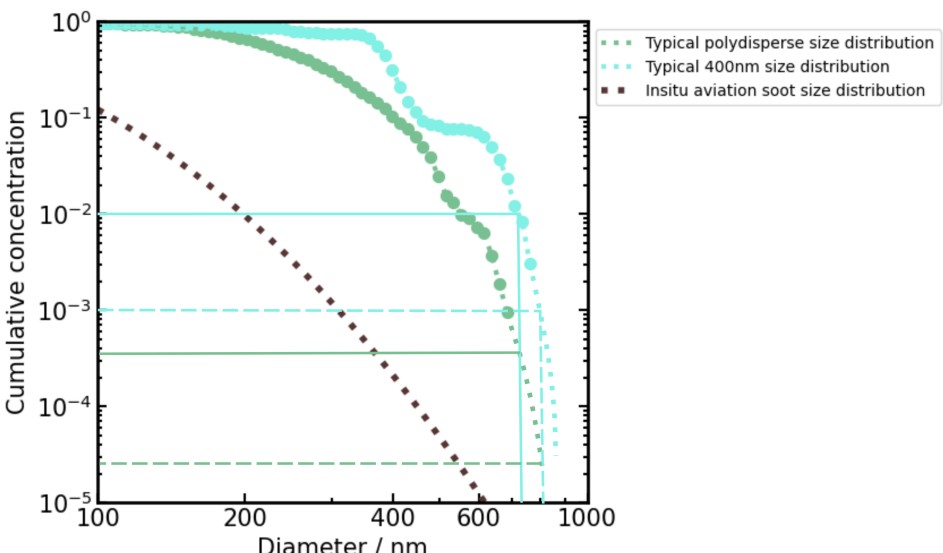

**Figure D1.** Cumulative concentration for typical 400 nm size-selected soot size distribution, polydisperse size distribution (mode diameter
around 250 nm) and insitu aviation soot size distribution. The largest (0.1 %) 1 % - i.e. $(10^{-3})$ $10^{-2}$ fractional concentration - particles from
the 400 nm size-selected size distribution have a diameter (> 800 nm) > 730 nm while this corresponds to only (0.0025 %) 0.035 % - i.e.
$(2.5 \times 10^{-5})$ $3.5 \times 10^{-4}$ fractional concentration - of the polydisperse aerosol population (shown with the solid and dashed lines).




**Appendix E: Single engine measured activated fractions for mono- and poly-disperse unCS- and CS-soot particles**

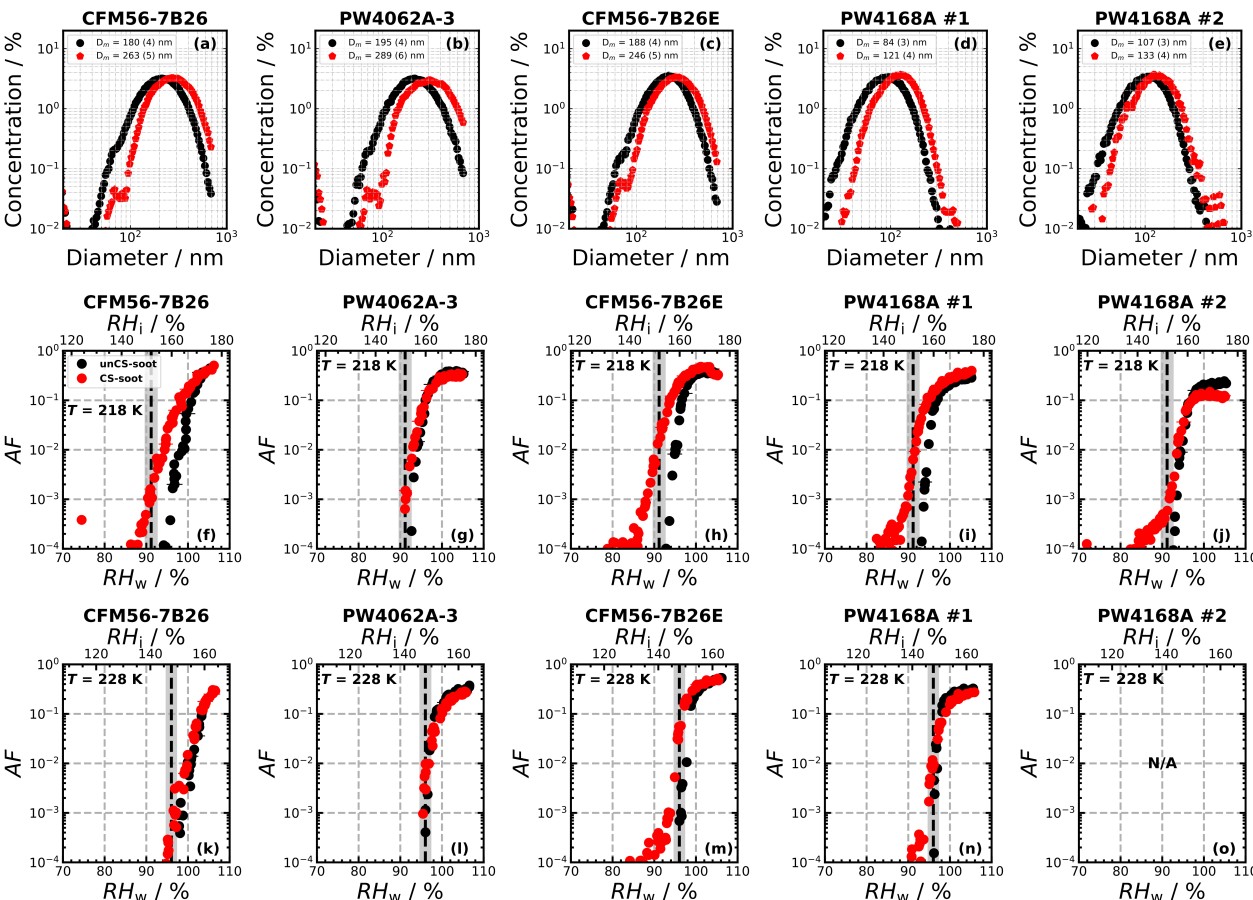

**Figure E1.** (a-e) Size distributions associated with the ice nucleation measurements shown on panels f-o. Uncertainties on the mode diameters are shown in parenthesis. (f-o) activated fraction (AF) as a function of RH for given temperatures and polysdisperse soot sampels from given engines. Each curve corresponds to a single experiment run. The dashed black lines show the RH at which aqueous solution droplets freeze homogeneously (Koop et al., 2000), and the grey shaded area represents uncertainties in RH across the aerosol lamina in the horizontal ice nucleation chamber (HINC).




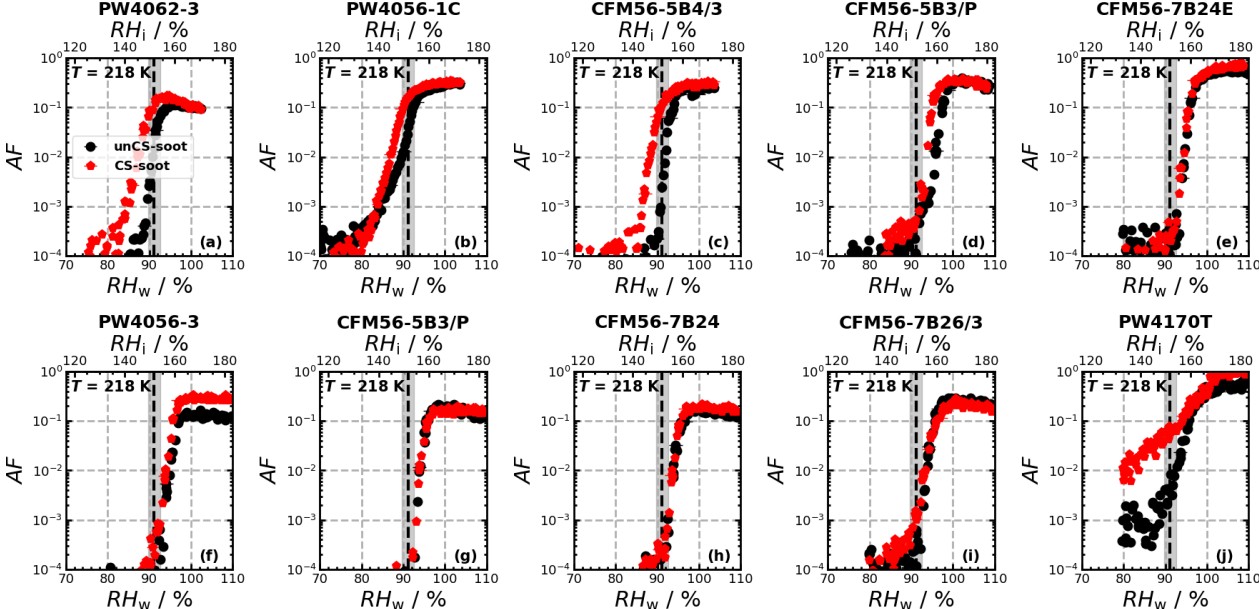

**Figure E2.** AF as a function of RH at 218 K for 400 nm size-selected aviation soot particles. Each curve corresponds to a single experiment run. The dashed black lines show the RH at which aqueous solution droplets freeze homogeneously (Koop et al., 2000), and the grey shaded area represents uncertainties in RH across the aerosol lamina in HINC.





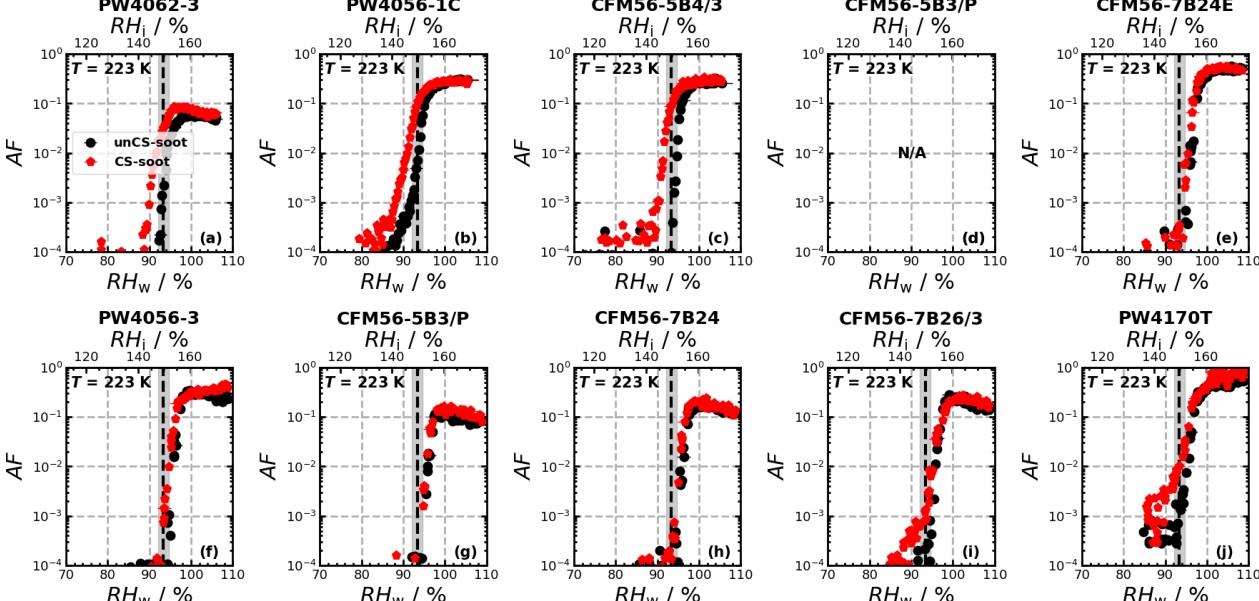

**Figure E3.** Same as Fig. E2 but at 223 K.



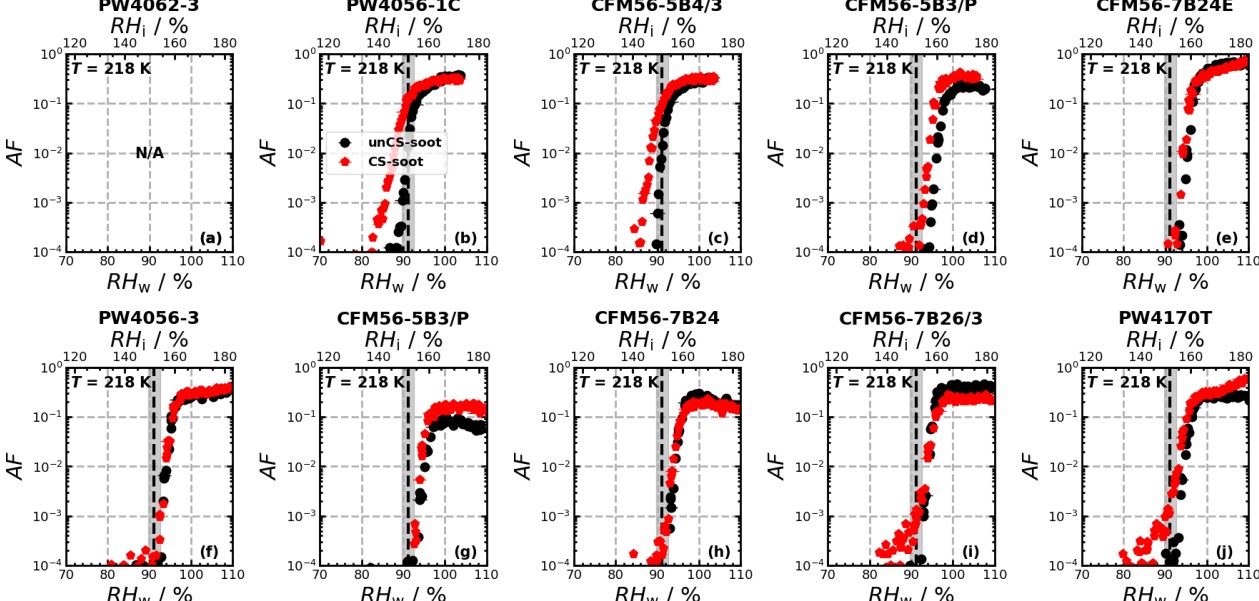

**Figure E4.** Same as Fig. E2 but for 200 nm soot at 218 K.



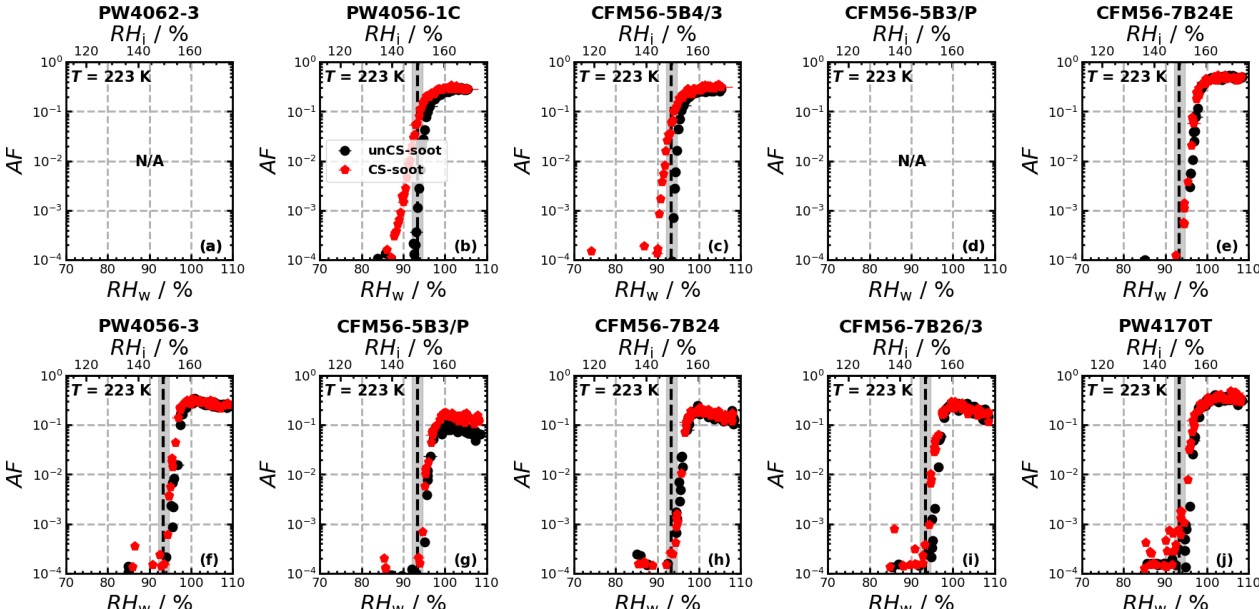

**Figure E5.** Same as Fig. E2 but for 200 nm soot at 223 K.




## Appendix F: Soot morphology analysis from TEM images

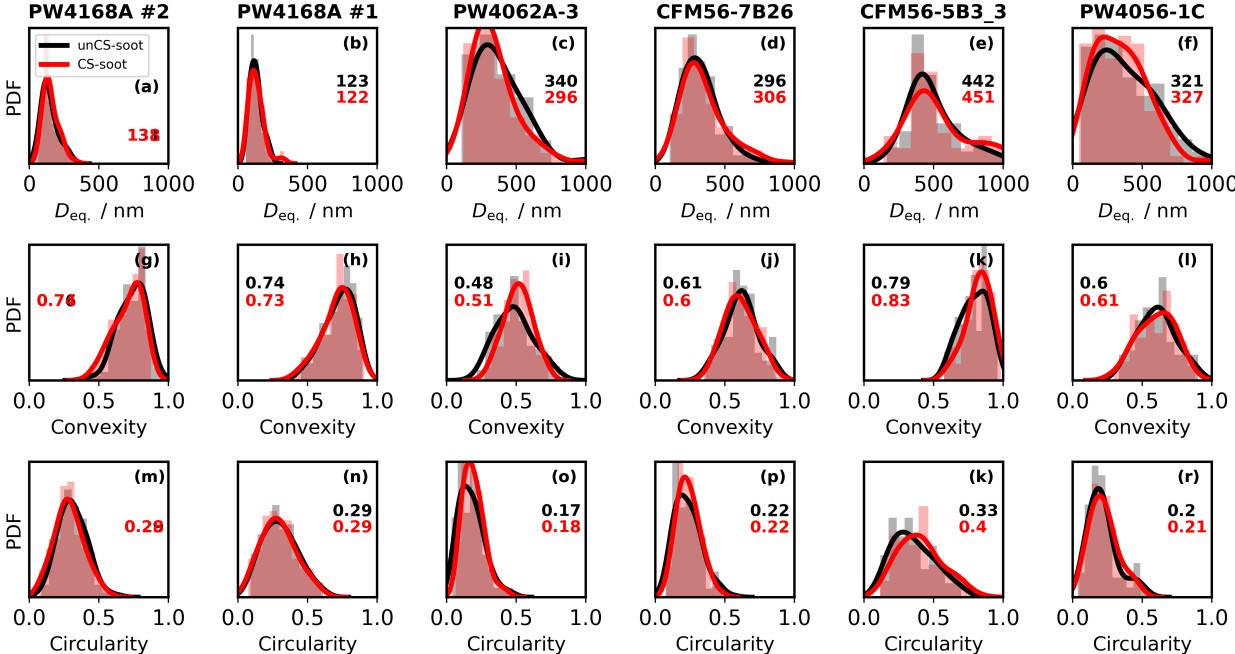

**Figure F1.** Probability density function (PDF) of the particles (a-f) spherical equivalent diameter ($D_{eq.}$, (g-l) convexity and (m-r) circularity. Median value of the distributions are displayed on the figures (in black for unCS-soot and in red for CS-soot). PDF are derived from about 100 to 140 transmission electron microscopy (TEM) images of soot aggregates for each unCS and CS samples (Table B1).



**Appendix G: Soot sample mass loss from Thermal gravitational analyser (TGA)**

Mass loss of of the aviation soot sample with the TGA amount to 10 % at 623 K. We further note that the mass loss during
in-line stripping of the 200 and 400 nm size-selected particles with the catalytic stripper led to generally smaller mass reduction
(except for the PW4170T engine, discussed in Sect. 3.4.2). Yet, polydisperse particles collected onto filter (heated in the TGA)
are smaller and the in-line mass measurements indicate that stripped mass increases with decreasing aggregate size, which is
coherent with increasing surface to volume ratio, i.e., increasing surface to mass ratio for smaller aggregates, and explain the
higher mass loss recorded in the TGA.

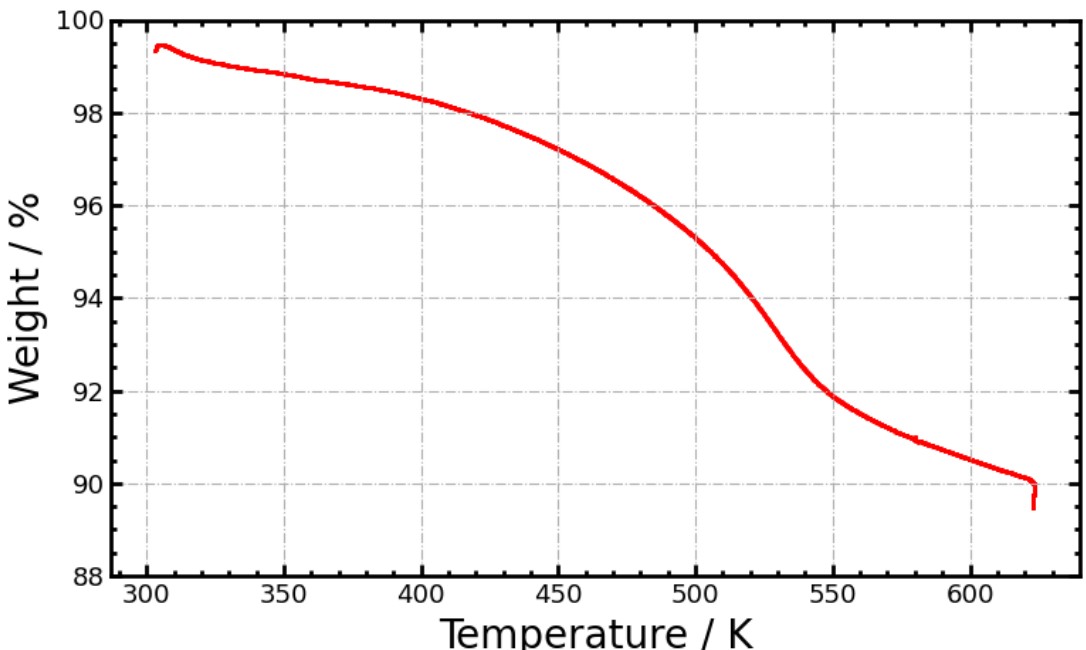

**Figure G1.** Mass loss recorded in the thermogravimetric analyzer (TGA) during heating of the mixed-thrust soot sample up to 623 K



**Appendix H: STXM/NEXFAS oxygen K-edge spectra of the soot samples**

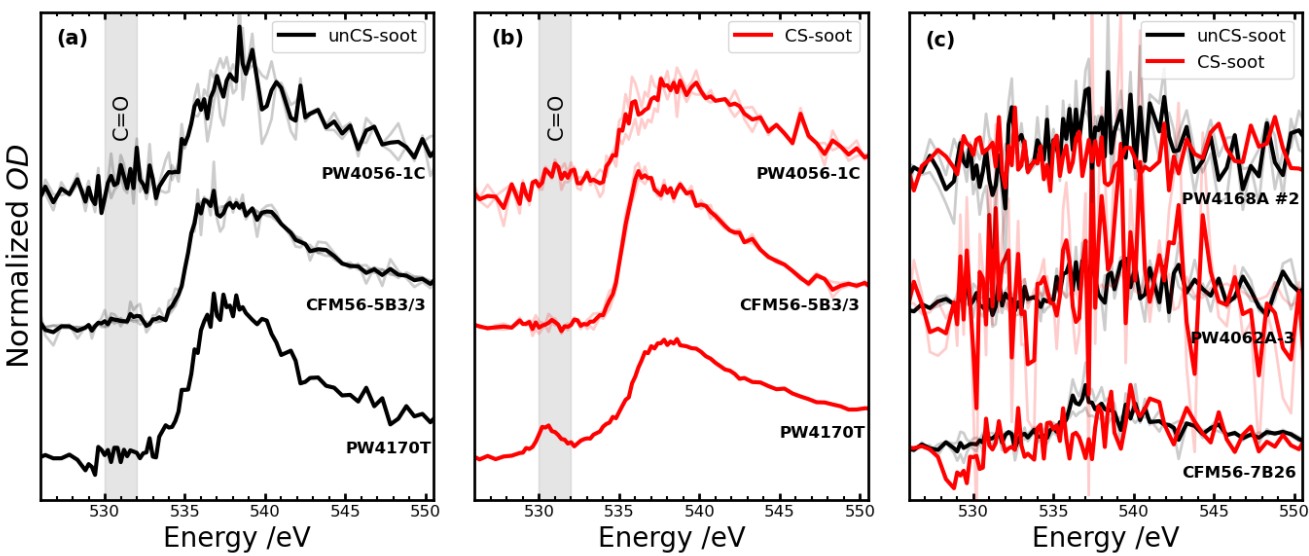

**Figure H1.** Near edge X-ray absorption fine structure (NEXAFS) spectra at the oxygen K-edge for unCS- and CS-soot of the given engines. The spectra for the different engines are offset for clarity. Light and bright lines correspond to single particle and mean spectra respectively. Spectra have been normalized for the total oxygen similarly as for the carbon K-edge (Sect. 2.5.2) with pre- and post-edge defined over 525-528 eV and 540-550 eV, respectively.



**Appendix I: STEM-EDX soot particles mapping for the PW4056-1C and CFM56-5B3/3 engines**



**Figure I1.** Panels a, e, i and m show dark field images of single unCS- and CS-soot aggregate emitted from the PW4056-1C and CFM56-5B3/3 engines. Associated scanning transmission electron microscopy-energy dispersive X-ray (STEM-EDX) mapping are shown for carbon, oxygen and sulfur.



**Appendix J:  Homogeneous ice nucleation rate coefficients**

The time needed for ice nucleation of the pore water depends on volume and on the homogeneous ice nucleation rate coefficient $J_{\mathrm{hom}}$ (Lohmann et al., 2016), the latter being a strong function of temperature (e.g. Ickes et al., 2015).

Using the parameterization from Ickes et al. (2015) and taking into account the effect of negative pressure (reached in the pore) on the nucleation rate (Marcolli, 2020) we can derive first order estimates of the homogeneous ice nucleation rate coefficient: For example, assuming a simple cylindrical pore of 2.5 nm diameter and 10 nm length with a contact angle of 70°, the homogeneous ice nucleation rate coefficient $J_{\mathrm{hom}}$ at 218 and 223 K equals $1.7 \times 10^{20}$ cm$^{-3}$ s$^{-1}$ and $3.2 \times 10^{19}$ cm$^{-3}$ s$^{-1}$, respectively. We further note that despite $J_{\mathrm{hom}}$ being 5 times higher at 218 K, both are high enough to promote freezing in

mesopores within the aerosol residence time in HINC.

However, $J_{\mathrm{hom}}$ also depends on the water activity of the aqueous solution which is nucleating. Figure J1 show $J_{\mathrm{hom}}$ at 218 and 223 K of a $H_2SO_4$ aqueous solution for varying $H_2SO_4$ mass fraction. The $H_2SO_4$ solution water activity has been computed from (Carslaw et al., 1995) and $J_{\mathrm{hom}}$ from (Koop et al., 2000). The latter is valid for water activity criterion between 0.26 and 0.34 corresponding to $H_2SO_4$ mass fraction of 13-20 wt % at 218 K and 10-19 % at 223 K.

The smallest pores with the smallest volume available for ice nucleation might no longer be able to form ice within the 10 s residence time in HINC due to the presence of $H_2SO_4$. Adding to this, the dependence of $J_{\mathrm{hom}}$ on temperature (Fig. J1) would allow less pore to nucleate ice at 223 K compared to 218 K resulting in a higher RH needed for ice nucleation onset as observed for 400 nm PW4056-1C in Fig. 6a and b.

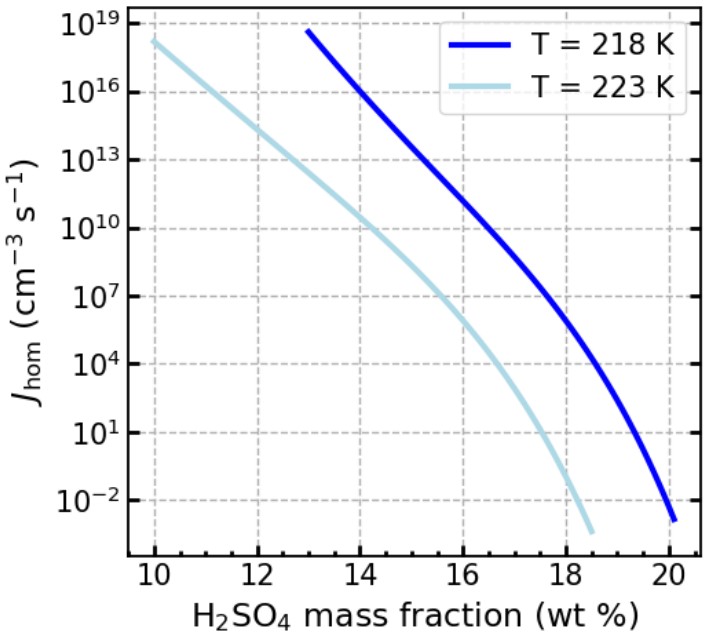

**Figure J1.** Homogeneous ice nucleation rate coefficient $J_{\mathrm{hom}}$ as function of $H_2SO_4$ mass fraction at given temperatures.



## Appendix K: Mass measurements of the aviation soot samples

**Table K1.** Soot particle mass change after stripping and mass mobility parameters derived from centrifugal particle mass analyser (CPMA) measurments. The prefactor $C$ is in $\mathrm{kg.m}^{-D_{\mathrm{fm}}}$ Number in parenthesis represent uncertainties on the measurements.

| Engine model | Particle mass change after stripping / % | | Mass–mobility exponent $D_{\mathrm{fm}}$[a] | Density–mobility prefactor $C$[a] | Mass–mobility exponent $D_{\mathrm{fm}}$[b] | Density–mobility prefactor $C$[b] |
|---|---|---|---|---|---|---|
| | 200 nm | 400 nm | | | | |
| PW4062-3 | – | – | – | – | – | – |
| CFM56-5B3/3 | – | – | – | – | – | – |
| PW4056-1C | – | – | – | – | – | – |
| CFM56-5B4/3 | – | – | – | – | – | – |
| CFM56-5B3/P | – | -5.6 (0.9) | – | – | – | – |
| CFM56-7B24E | -4.2 (1.1) | -3.8 (0.6) | – | – | – | – |
| PW4056-3 | +0.4 (0.2) | – | 1.85 (0.11) | $9.8 \times 10^{-6}$ ($1.6 \times 10^{-6}$) | 1.71 (0.22) | $1.1 \times 10^{-6}$ ($3.7 \times 10^{-6}$) |
| CFM56-5B3/P | – | +1.7 (1.8) | 1.82 (0.09) | $7.8 \times 10^{-6}$ ($1.0 \times 10^{-5}$) | 1.68 (0.19) | $1.1 \times 10^{-6}$ ($3. \times 10^{-6}$) |
| CFM56-7B24 | -2.8 (1.8) | -1.4 (0.6) | 1.76 (0.07) | $3.5 \times 10^{-6}$ ($3.9 \times 10^{-6}$) | 1.62 (0.14) | $4.8 \times 10^{-7}$ ($1.0 \times 10^{-6}$) |
| CFM56-7B26/3 | -3.0 (1.7) | -2.4 (0.5) | 1.65 (0.09) | $6.9 \times 10^{-7}$ ($9.3 \times 10^{-7}$) | 1.66 (0.15) | $8.1 \times 10^{-7}$ ($1.8 \times 10^{-6}$) |
| PW4170T | -10.3 (2.5) | – | – | – | 1.88 (0.14) | $1.7 \times 10^{-5}$ ($3.6 \times 10^{-5}$) |
| CFM56-7B26 | – | – | – | – | – | – |
| PW4062A-3 | – | – | – | – | – | – |
| CFM56-7B26E | – | – | – | – | - | – |
| PW4168A #1 | – | – | – | – | – | – |
| PW4168A #2 | – | – | – | – | – | – |

[a] mass-mobility law fit over 100-600 nm particles, [b] mass-mobility law fit over 200-600 nm particles.



*Author contributions.* BT and ZAK designed the experiments with help from LD, JA, JE and CS. BT conducted the experiments and performed the data analysis. The STXM/NEXAFS measurements were conducted by BT, PA, FM and ZD. The DVS measurements were performed by CD. BT wrote the first draft and all authors contributed to data interpretation and writing of the manuscript. ZAK supervised the study, conceived the idea and obtained funding.

*Competing interests.* The authors declare that they have no conflict of interest.

*Acknowledgements.* This work was carried out under the European Commission via their Horizon 2020 Research and Innovation Program under Grant Number 875036 (ACACIA project) and the Swiss Federal Office of Civil Aviation, SFLV-2020-080. We also acknowledge funding from the European Union's Horizon 2020 research and innovation programme under the Marie Skłodowska-Curie grant agreement no. 890200. The authors are grateful for the support and help of the engine operators of the SRT facility. We also thank Kunfeng Gao (LAPI-EPFL) and Cuiqi Zhang (IAC-ETH) for the helpful discussions and comments on the manuscript. The authors are greatfull to Benjamin
Watts and Simone Finizio from the Swiss Light Source (PSI) for their support with STXM/NEXAFS measurements. The authors further thank Kevin Kilchhofer from Laboratory of Atmospheric Chemistry (PSI) for helping with the STXM/NEXAFS measurements. We also thank Prof. Pratsinis from the Departement of Mechanical and Process Engineering (ETH) for lending a CPC during our measurement campaigns.



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
