# Peer review of "Soot aerosol from commercial aviation engines are poor ice nucleating particles at cirrus cloud temperatures"

_EGUsphere, 2023_

## Author Comment (AC1)

Response to egusphere-2023-2441 reviews for RC1

**Referee comments are marked in black bold and are numbered as R1Cx with x the comment number.** Author (AC) responses are marked in black directly below the comments. The original text from the manuscript is repeated in blue and corrected text in revised manuscript is typed in red. Previous line numbers given in "()" following the updated line numbers.

**The authors Testa et al describe a series of ice nucleation measurements conducted on authentic aircraft engine soot and accompanied by chemical composition and morphology analyses for relation to INA. The extensive chemical and physical characterization of aviation soot support plausible explanations for observed INA and the results are important in the contexts of parameterizing atmospheric INA and understanding the many prior studies on soot and soot proxy INA. I have two comments/questions about the methodology and representativeness of the results and a few more minor comments below and would likely suggest publication after these comments are addressed.**

We thank Reviewer 1 for their time and comments and respond to the concerns individually below.

**R1C1: The authors analyze soot produced by P&W and CFM International engines using Jet-A1 type fuel. Approximately how large is the market/usage share for these engines and fuel type?**

P&W and CFM International engines represented 68 % of the aircraft engine fleet in 2020 (52 % for CFM engines and 16 % for P&W engines; FlightGlobal.com, 2021). Today, Jet A-1 and jet A fuel represent the major part of fuel consumption with jet A largely used in North America and Jet A-1 in the rest of the world (~30 % and >60 %, respectively; Jing et al., 2022; Pires, 2018), while SAF represents only 0.2% of global production (IATA, 2023). We note that Jet A-1 and Jet A have very close chemical composition (Pires, 2018) and presumably results in aviation soot with similar properties.

Following the comment, lines 83-85 (79) read as follow: "Sampling took place from several engines from Pratt and Whitney (P&W) and CFM International (16 % and 52 % of global market share in 2020, respectively; FlightGlobal.com, 2021), all fueled with Jet-A1 fuel (> 60 % of global usage; Jing et al., 2022; Pires, 2018)"

**R1C2: Can the authors comment on how generalizable the INA trends might be to other engines or fuel types?**

The INA of aviation soot depends primarily on the presence of mesopores in soot aggregate to promote ice nucleation via PCF. Relevant soot properties that affect the mesopore availability and hence their ice nucleation ability are the aggregate morphology and size. Aggregate morphology refers to how fused the primary particles are and the range of primary particle diameter within an aggregate. The aggregate size scales with the number of primary particles per aggregate. In addition, water-soluble/water-insoluble coatings (e.g., sulfur, organics) and soot surface functional groups also impact aviation soot INA.

P&W and CFM engines have different design but both engine types emit soot with highly fused primary particles. Similar features have been observed for an IAE V2527 engine (Vander Wal et al., 2022; manufactured by IAE International Aero Engines and designed namely by P&W and Rolls-Royce). However, measurements of aviation soot primary particle properties are scarce, making it difficult for a sophisticated, quantitative comparison with our results. The aggregate sizes for other engine, e.g., Rolls-

Royce and General Electric engines (respectively 7 % and 6 % of the global fleet; FlightGlobal.com, 2021) are in the same size range as for P&W and CFM engines (Lobo et al., 2015; Wahl & Aigner, 2003). We expect the presence of sulfur to inhibit soot pore ice nucleation for other engine types, if emitted sulfur associated with soot is in amounts comparable to this study. We acknowledge however that the soot sulfur content might vary for different engine types, as observed in our study. Overall, the above suggests that extrapolation of our results to other engine types operated with Jet A1/A fuel (> 90 % jet fuel production) is likely, in particular considering the small soot sizes (hence limited number of pores). Nonetheless, to be completely quantitative, dedicated measurements of other soot properties for different engine types would be needed.

Regarding the generalization to bio- and synthetic fuels, whose production is largely below that for jet A1/A fuel to date (R1C1): to the best of our knowledge, small to no difference in aggregate size and primary particle size has been found for alternative jet fuel soot (Huang & Vander Wal, 2013; Lobo et al., 2016; Vander Wal et al., 2022) but differences in particle surface properties (Liati et al., 2019). More such studies would be needed to be conclusive on the INA of soot emitted from alternative fuels.

**R1C3: The soot particles used for INA analysis undergo coagulation prior to analysis and the authors extrapolate results from coagulated particles to smaller particles that are more likely to be present in the atmosphere. Based on the TEM analysis, are there any differences in overall particle morphologies between uncoagulated and coagulated soot that may affect INA?**

The soot shape analysis from the TEM images were conducted for the coagulated soot only. However, primary particle properties (overlap, size, and oxidation) are fixed in the combustor (Dakhel et al., 2007), hence we expect those properties to be the same for uncoagulated and coagulated soot. From size distribution measurements (Fig. D1) we know that the coagulated soot particles are larger with corresponding mass measurements (Table K1) showing that larger particles have a lower fractal dimension and hence are lacier. This is also verified with the TEM images of the coagulated soot; smaller soot particles are generally more convex (= compact) than larger ones. This is visible when plotting particle convexity as a function of their $D_{eq}$ retrieved with TEM (the data shown as separate distributions on Fig. F1 are plotted against each other in Figure 1 at the end of the present document). Compaction of particles generally increases the INA (Mahrt et al., 2020). Yet, as uncoagulated soot have much smaller sizes than coagulated soot and hence a limited number of primary particles, the probability for uncoagulated soot to have PCF relevant pores is small. As such, we expect the INA of uncoagulated soot particles to be lower than that of coagulated soot particles (as stated in the paper lines 508-511). The size dependency of soot particles has been shown in numerous studies before, where for example 100 nm particles are poor at ice nucleation compared to 400 nm particles of the same sample (K. Gao et al., 2022; Mahrt et al., 2018; Zhang et al., 2020).

**R1C4: The authors discuss that compaction of soot particle aggregates may decrease INA (lines 412-420). Based on the TEM images in Figures C1 a-e and k-o, there appear to be some differences in particle morphology (e.g., fractal dimension) between uncoagulated and coagulated particles. Can the authors comment on any potential differences in particle morphology and INA as a result of coagulation?**

In lines 425-433 (412-420) we compare coagulated soot from the CFM56-5B3/3 and PW4056-1C engines and state that CFM56-5B3/3 soot appears very compact (more than PW4056-1C soot). High soot compaction can result from a very fused primary particles network, limiting room for cavities between the primary particles, which explains its poorer INA compared to the PW4056-1C. In other words, we postulate that compaction leads to decreased INA in the case of highly fused primary particles, as for the CFM56-5B3/3 soot discussed here.

We acknowledge that there appear to be differences between the uncoagulated and coagulated soot shown in Fig. C1 a-e and k-o, namely soot shown on Fig. C1 k-o (CFM56-5B3/3) being very compact. We note that despite their large aggregate sizes, CFM56-5B3/3 soot are very compact in comparison to all other engine tested (high and narrow convexity, Fig. F1). Soot from this engine does not follow the near linear relation between aggregate size and convexity verified for most engine types tested in this study (Figure 1 at the end of this document). Hence, we hypothesized that the compactness of the coagulated CFM56-5B3/3 soot originates rather from the engine type than the coagulation process. The overall morphology difference between uncoagulated and coagulated soot and impact on INA are already discussed in R1C3.

**R1C5: Can soot morphology change during impaction and could this vary with particle size?**

Soot aggregates can (partially) break apart upon impaction (Gao & Kanji, 2022), especially if the bonds between soot monomers are fragile due to prior heating (as for our CS-soot samples). We however did not detect any soot aggregate fragments with sizes similar to individual primary particles on the TEM images that would indicate aggregate break-up. We also did not observe a morphology difference between unCS- and CS-soot, ruling out fragmentation of the heated particles upon impaction. Other possible effects are flattening of the particles, possibly more pronounced for large aggregates or bouncing of the particle on the grid, causing structural change (Virtanen et al., 2010). Such effects are likely limited with the Partector TEM sampler (4 out of 6 of samples) using electrostatic softer (than aerodynamic) impaction.

Second, multiple soot particles might aggregate on the TEM grid if impacting at the same position. This can happen with our aerodynamic based TEM sampler (ZEMI) but is limited for samples collected with the Partector TEM sampler as this technique uses electrostatic impaction. Nonetheless, we avoided imaging particles that would be the result of aggregation on the grid upon compaction by carefully imaging clearly isolated soot aggregates.

**R1C6: Line 40: these are cavities and voids formed by soot spherule overlap, compaction, and aggregation, correct? Clarify this, to make clear that cavities aren't present on the surfaces of individual spherules.**

**R1C7: Line 41: clarify that this initial freezing is step two in the list presented here.**

This is correct.

Lines 40-41 (40-41) now read: "In a first step, water vapor condenses in soot aggregate pores (i.e., cavities and voids between overlapping and aggregated soot primary particles) below bulk water saturation $RH_w <$ 100 %, followed by a second step where pore water freezes homogeneously at $T < T_{hom}$"

**R1C8: Line 44: approximate size of the critical ice cluster?**

From the CNT parameterization from Ickes et al. (2015) and Murray et al. (2010), Marcolli et al. (2021) estimated the size of the critical ice cluster to range from 0.5 to 1.5 nm for 200 K < $T$ < 235 K.

Lines 45-46 (44) now read: "[…] to pores that can accommodate the critical ice cluster (David et al., 2020) with sizes of 0.5-1.5 nm for 200 K < $T$ < 235 K (Marcolli et al., 2021)."

**R1C9: Line 47: clarify "overlap:" the extent to which primary spherules are "pressed" together by sintering and compaction?**

Lines 48-51 (47) now read: "[…] their degree of overlap (Brasil et al., 1999; Marcolli et al., 2021). Primary particle overlap represents the extent to which spherical soot primary particles are fused due to surface growth from soot precursor following primary particle coagulation, and pressed together by sintering and compaction. By definition, primary particles in point contact are not overlapped."

**R1C10: Line 56: is the sulfur internally mixed with soot present as sulfuric acid? This line implies that it is but is not explicitly clear.**

Sulfur has been reported to be internally mixed with soot on the form of $H_2SO_4$ coatings (Kärcher et al., 1998; Kärcher et al., 2007; Schumann et al., 2002), but can also be associated with carbon (Popovicheva et al., 2004), like in organic thiophene or thiophenol (Parent et al., 2016).

Now lines 59-60 (56) read: "As an example, sulfur compounds present in aviation fuel (about 500-600 ppm for Jet A-1 fuel; Lee et al., 2021; Starik, 2007) can be internally mixed with aviation soot as associated with carbon (Parent et al., 2016; Popovicheva et al., 2004) and in the form of $H_2SO_4$ (Kärcher et al., 1998; Kärcher et al., 2007; Schumann et al., 2002)."

**R1C11: Lines 64-65: I'm a little confused by the phrase "averaged aggregate $D_{PP}$." From the context, it sounds like this sentence is saying that $D_{PP}$ increases in size with increasing thrust, but I'm not certain what the modifier "averaged aggregate" means for $D_{PP}$.**

The denomination "averaged aggregate $D_{pp}$" was chosen because values for $D_{pp}$ over single aggregate can be variable (e.g., from 10-100 nm; Vander Wal et al., 2022), but the averaged or median aggregate $D_{pp}$ increase with thrust. Because the aggregate $D_{pp}$ range is introduced only introduced later in the results sections and to avoid confusion, we propose the following simplification:

Line 68-69 (64-65) modified: "Similarly, $D_{pp}$ has been shown to increase with thrust, potentially resulting in different soot porosity at different engine thrust."

**R1C12: Line 98: could the size-dependency of particle losses impact the INA measurements in this work? Might particles losses be significant for this analysis given the detection limits of the HINC and low INA of the samples?**

Particle loss (by coagulation and diffusion) can be significant between the engine probe and the aerosol reservoir as particle concentration there is high (about $10^7$ cm$^{-3}$). However, downstream of the tank, particle concentration was reduced to about 100-500 cm$^{-3}$ with a dilution system for the polydisperse measurements and by the DMA for the size selected measurements. What matters for the soot INA would be particles losses between the CPC/SMPS and HINC. We expect this to be limited due to low flow in the tubes (limiting inertial losses), the low particle concentration (100-500 cm$^{-3}$, limiting losses by particle coagulation). The short and straight tubing and the particle sizes, i.e., limited small (< 10 nm) and large (> 100 nm) particles, limit diffusional and inertial/gravitational losses (Brockmann, 2011). Line 103 (98) was referring to particle loss that might be important between the engine probe and the aerosol reservoir as particle concentration in this line is high ($10^7$ cm$^{-3}$) but that we do not quantify as the study does not aim at quantifying engine emissions.

**R1C13: Line 105: to what extent is rapid coagulation of aircraft aviation soot is expected under ambient conditions?**

Airborne in situ measurement of soot sizes in young aircraft plumes are scarce. Petzold et al. (1999); Petzold et al. (1998) and Twohy and Gandrud (1998) observed interstitial and contrail residual soot aggregates of 0.15-1 μm. Those large aggregates could result from the coagulation of soot aggregates trapped in the wingtip vortices (Miake-Lye R. C. et al., 1993) due to the higher soot emission index for older engine models (Lee et al., 2010; Masiol & Harrison, 2014). Coagulation of contrail ice crystals and merging of embedded soot aggregates upon sublimation of the ice crystals could also lead to larger soot aggregates. Yet, for current aircraft engines with lower emission indices, coagulation of the soot aggregate is reduced or inhibited, as shown by Moore et al. (2017) that did not observe soot coagulation, and whose measurements are similar to what would be measured at ground in engine maintenance and testing facility (e.g., Durdina et al., 2021). Thus, under ambient conditions, coagulation is likely not expected. In our study, soot particles undergo artificial coagulation in the aerosol reservoir, and we discuss the implication/limitation of this in the Atmospheric implication section.

**R1C14: Line 157: please include a description of how equivalent spherical diameter, convexity, and circularity are calculated and/or defined.**

In lines 162-166 (158), we have added: "The resulting images were used for morphology analysis by image and processing with the MATLAB (R2020a, MathWorks Inc., Natick, USA) code of Dastanpour and Rogak (2014), which was modified in order to retrieve the projected area equivalent diameter ($D_{eq}$) defined as $D_{eq} = \sqrt{\frac{4A_a}{\pi}}$, with $A_a$ the aggregate projected aggregate area; the convexity, defined as convexity $= \frac{A_a}{A_{convex}}$ with $A_{convex}$ the area of the smallest convex polygon enclosing the aggregate; and circularity, as circularity $= \frac{r_{ins}}{r_{circ}}$ with $r_{ins}$ and $r_{circ}$ the radii of circles inscribing and circumscribing the soot aggregate (China et al., 2015; Mahrt et al., 2020). Convexity and circularity are indicators of how round and compact aggregates are.

**R1C15: Line 190: given the spot size, each measurement is expected to originate from an individual particle? This is referenced later but should be clarified here.**

We agree. Lines 200-201 (190) now reads: "Spatial resolution was below 100 nm, resolving single soot aggregates, i.e., an individual spectrum corresponds to a single particle, with an energy resolution of 0.1 eV at the carbon and about 1.3-1.6 eV at the oxygen K-edges sufficient to identify any peak at both edges."

**R1C16: Line 267: how many experiments, exactly?**

Line 277 (267) now reads: "[…] with 3 out 6 experiments promoting ice nucleation at RH < RH$_{hom}$ for 400 nm aggregates at 218 K."

**R1C17: Line 323: please clarify the meaning of the first sentence of this paragraph.**

Lines 333-335 (323) and beyond now read: "Discriminating the CS-soot ice nucleation ability by engine type shows a clear discernable onset RH between P&W and CFM56 engines. This outcome is similar when we discern by thrust, where different thrust regimes lead to differentiable RH onset. Yet, as for the unCS-soot, due to the thrust confounding factor (3 out of the 4 P&W engines run at mixed-thrust), the engine type effect on the ice nucleation of CS-soot is not elucidated. For high-thrust conditions […]"

We also clarify in lines 323-325 (313-314): "The median ice nucleation onset for CS-soot is at or above RH$_{hom}$ at all experimental conditions, comparable to unCS-soot ("All engine experiments" in Fig. 3).

However, the onsets for CS-soot spread to a wider RH$_i$ range compared to unCS-soot, and if we consider the effect […]"

**R1C18: Line 372: Can sulfur be present in samples in forms other than sulfuric acid, for example bound to carbon? What sulfur bonding environment presents an overlap at the 538 eV value? Are there any signals clearly attributable to sulfur or is the expected signal (based on the atomic %, line 405) below the limit of detection?**

As mentioned in R1C10, sulfur on soot has been observed as associated with carbon (Popovicheva et al., 2004), e.g., like in organic thiophene or thiophenol (Parent et al., 2016). The relative fraction of organic sulfate versus H$_2$SO$_4$ cannot be inferred from our measurement. However, sulfur in soot pores that might contain condensed water is expected to be found in the form of ionic sulfate (HSO$_4^-$ and SO$_4^{2-}$) and H$_2$SO$_4$ (although water is thought to evaporate in the STXM and TEM vacuum).

Sulfur K- and L-edge are around 2470 and 160 eV respectively (Jalilehvand, 2006) but those energies were not accessible with the STXM device that we used. However, oxygen associated with sulfur as S-O bonding like in ionic sulfate and H$_2$SO$_4$ should contribute to absorption at 538 eV (Zelenay et al., 2011) for the unCS-soot sample. The concentration of oxygen and sulfur for the PW4056-1C and CFM56-5B4/3 engines are about 0.11 and 0.26 atomic % for sulfur and 4 and 9 atomic % for oxygen. Assuming that most sulfur is present as ionic sulfate and H$_2$SO$_4$, one sulfur atom is bonded to 2 oxygen with a single bond, hence atomic fraction of oxygen in S-O represents about 0.06% (2 x 0.11/4 and 2 x 0.26/9) of total oxygen for both engines. Hence, oxygen in S-O represents only as small contribution to absorption compared to C=O and C-O for the unCS-soot sample (and do not contribute to the CS-soot sample).

Text has been added following lines 385-386 (374): "Sulfur should also contribute to the oxygen spectra at 538 eV (Zelenay et al., 2011), but only for unCS-soot particles as CS-samples should be depleted of sulfur, as observed in the STEM-EDX measurements (Fig. 5c). The sulfur contribution to oxygen spectra is however expected to be insignificant given the low amount of sulfur, hence low amount of oxygen bound to sulfur (Fig. 5c)"

**R1C19: Line 405: are the H2SO4 wt % in Figure J1 relevant to the expected amount of sulfuric acid and water based on the sulfur atomic %?**

In order to estimate H$_2$SO$_4$ wt % in an aggregate pore, the total H$_2$SO$_4$ wt % per aggregate needs to be computed. From our measurement, this can be retrieved from sulfur atomic % measured with STEM-EDX. For instance, for PW4056-1C soot: sulfur atomic % is 0.11 (Fig. 5c). The conversion to wt % is computed as follows:

$$\text{wt } \% \ x = \frac{\text{atomic } \% \ x \times \text{atomic wt } x}{\sum \text{atomic } \% \ i \times \text{atomic wt } i} \qquad (1)$$

With $i$ being the elements present on the sample. Assuming that only C, O and S are present on the aggregate (i.e., ignoring hydrogen that cannot be quantified with STEM-EDX and other trace components), with S atomic % = 0.11 %, O atomic % = 4 % and C atomic % = 100 − 4 − 0.11 % (Fig. 5C),  from equation 1:

sulfur wt % $= \frac{0.11 \times 32}{0.11 \times 32 + (100-4-0.11) \times 14 + 4 \times 16} \times 100 = 0.28$ wt %. There are 4 oxygen atoms associated with the sulfur atoms in H$_2$SO$_4$, hence the atomic % of oxygen associated with H$_2$SO$_4$ in the soot aggregates

is 4 x sulfur atomic % = 4 x 0.11 atomic % = 0.44 atomic %. Then again, using equation 1: total oxygen wt % $= \frac{4 \times 16}{0.11 \times 32+(100-4-0.11) \times 14+4 \times 16} \times 100 = 5.25$ wt %, which yields to oxygen wt % in $H_2SO_4$ = $\frac{5.25 \text{ wt \%} \cdot 0.44 \text{ atomic \%}}{4 \text{ atomic \%}} = 0.58$ wt %. Again, ignoring the mass fraction of hydrogen in $H_2SO_4$, the $H_2SO_4$ wt % = 0.58 + 0.28 = 0.86 wt %, on average over soot aggregates for the PW4056-1C engine.

Next, pore volume per aggregate needs to be computed. This can be retrieved from the desorption isotherm measured with the DVS (Fig. 4). The method has been detailed in Mahrt et al. (2020) and includes several assumptions that are worth mentioning. One assumes that the adsorption of water is solely due to pore capillary condensation, i.e., no water uptake due to soluble coating at the surface and that the pores are cylindrical. Additionally, a single value of aggregate contact angle needs to be assumed. Then pore volume $V_p$ per aggregate mass [cm$^3$ g$^{-1}$] is computed by integrating over given pore radius ($r$), i.e., $V_p = \int_{r2}^{r1} \frac{dV_p}{dr} dr$.

It follows that the volume fraction of $H_2SO_4$ in a pore is:

$$H_2SO_4 \text{ vol \%} = \frac{V_{H2SO4,agg}}{V_{p,agg}} = \frac{\frac{m_{H2SO4,agg}}{\rho_{H2SO4}}}{V_p \cdot m_{agg}} = \frac{\frac{\text{wt \%}_{H2SO4,agg} \cdot m_{agg}}{\rho_{H2SO4}}}{V_p \cdot m_{agg}} = \frac{\text{wt \%}_{H2SO4,agg}}{V_p \cdot \rho_{H2SO4}}$$

with $V_{H2SO4, agg}$ [cm$^3$] and $m_{H2SO4, agg}$ [g] the volume and mass of $H_2SO_4$ per aggregate, respectively; $V_{p, agg}$ [cm$^3$] the pore volume per aggregate; $\rho_{H2SO4}$ = 1.0735 [g cm$^{-3}$] the density of $H_2SO_4$ in aqueous solution (assuming a solution with H2SO4 wt % = 10 % at 0 °C; Green & Perry, 2007). $m_{agg}$ [g] is the aggregate mass and finally wt % $_{H2SO4, agg}$ the $H_2SO_4$ mass fraction per aggregate calculated above.

Considering the desorption branch of the unCS mixed-thrust soot (Fig. 4), integrating $V_p$ over the entire pore size range (0.1 to 10 nm for the PW4056-1C soot), and assuming a contact angle of 70°C (65° - 80° suggested for aviation soot proxy in Kunfeng Gao et al., 2022), this yields an $H_2SO_4$ vol % = 1.32 %. Assuming then the pores are entirely filled with water, it follows $H_2SO_4$ wt % in pores = 1.41 wt %. As a comparison, if we assume that only pore < 3 nm are filled (assuming for instance that $H_2SO_4$ condense is the smallest pore first due to their lower equilibrium vapor pressure) this yield to $H_2SO_4$ vol % in pores = 3.32 vol % = 3.45 wt %.

Those values are below the 10-20 wt % range assumed in Fig. J1. Yet, those need to be considered very carefully given the numerous assumptions made in the above. Given that $H_2SO_4$ is thought to be present at the surface, it should contribute to water uptake at low RH. Yet, the portion of the isotherm at low RH determine the volume of micropores ($r$ < 2 nm), hence the assumption that only micropore contribute to water uptake at low RH leads to overestimation of their volume. In the derived pore volume distribution, micropores represent 30 % of the total pore volume, reducing the overall $H_2SO_4$ mass fraction in mesopores. Next, the content of sulfur retrieved from EDX is likely underestimated due to sulfate evaporation in the microscope vacuum. Moreover, soot pores, e.g., ring pores, are not cylindrical by rather wedge-shaped (Marcolli et al., 2021), and are likely not described by single contact angle value.

All in all, given the large uncertainty associated with the derived pore volume distribution and $H_2SO_4$ content, one cannot be conclusive of the mass fraction of $H_2SO_4$ in aviation soot pores. Targeted measurements would be needed to give representative estimation. Nonetheless, the computed concentration is in the order of magnitude of concentration assumed in Fig. J1 and likely contributes to lowering the pore water activity, inhibiting ice nucleation.

**R1C20: Lines 515-516: is this assumption on real-world coating extent based on SOA and sulfate that condenses on particles after emission, or engine emissions condensing onto soot due to low temperatures? The ambient atmosphere is also a much more dilute environment than the chamber, which would disfavor thicker coatings.**

We refer here to the interaction of soot with aircraft condensable vapors (mainly sulfur and organic vapor) and with nucleation mode particles ($H_2SO_4$ and organic oil droplets) in the young aircraft plume, suggested by multiple studies, despite some conducted at ground level, where ambient temperature and dilution dynamic might differ from high altitude plume (Kärcher et al., 2007; Onasch et al., 2009; Ungeheuer et al., 2022; Wong et al., 2014; Yu et al., 2014).

Besides, aviation soot particles can also interact with ambient aerosols (SOA, sulfate) and semi-volatiles during the residence time in the upper troposphere, i.e. days to weeks (Bond et al., 2013), adding to the potential of reducing INA due to soot pore blocking.

**R1C21: Figure C1: please add length scale labels that are more legible.**

Figure C1 has been modified.

[Figure]

Figure 1: unCS-soot (in black) and CS-soot (in red) aggregate convexity as function of $D_{eq}$ for the given engines. Convexity and $D_{eq}$ density distributions are shown in the manuscript on Fig. F1 for the same engines.

Bond, T. C., Doherty, S. J., Fahey, D. W., Forster, P. M., Berntsen, T., DeAngelo, B. J., Flanner, M. G., Ghan, S., Kärcher, B., Koch, D., Kinne, S., Kondo, Y., Quinn, P. K., Sarofim, M. C., Schultz, M. G., Schulz, M., Venkataraman, C., Zhang, H., Zhang, S., . . . Zender, C. S. (2013). Bounding the role of black carbon in the climate system: A scientific assessment. *Journal of Geophysical Research: Atmospheres*, 118(11), 5380-5552. https://doi.org/https://doi.org/10.1002/jgrd.50171

Brockmann, J. E. (2011). Aerosol Transport in Sampling Lines and Inlets. In *Aerosol Measurement* (pp. 68-105). https://doi.org/https://doi.org/10.1002/9781118001684.ch6

Dakhel, P. M., Lukachko, S. P., Waitz, I. A., Miake-Lye, R. C., & Brown, R. C. (2007). Postcombustion Evolution of Soot Properties in an Aircraft Engine. *Journal of Propulsion and Power*, 23(5), 942-948. https://doi.org/10.2514/1.26738

David, R. O., Fahrni, J., Marcolli, C., Mahrt, F., Brühwiler, D., & Kanji, Z. A. (2020). The role of contact angle and pore width on pore condensation and freezing. *Atmospheric Chemistry and Physics*, 20(15), 9419-9440. https://doi.org/10.5194/acp-20-9419-2020

Durdina, L., Brem, B. T., Elser, M., Schönenberger, D., Siegerist, F., & Anet, J. G. (2021). Reduction of Nonvolatile Particulate Matter Emissions of a Commercial Turbofan Engine at the Ground Level from the Use of a Sustainable Aviation Fuel Blend. *Environmental Science & Technology*, 55(21), 14576-14585. https://doi.org/10.1021/acs.est.1c04744

FlightGlobal.com. (2021). *Commercial engines*. https://www.flightglobal.com/reports/commercial-engines-2021/143946.article, last accessed on 01.2024

Gao, K., Friebel, F., Zhou, C. W., & Kanji, Z. A. (2022). Enhanced soot particle ice nucleation ability induced by aggregate compaction and densification. *Atmospheric Chemistry and Physics*, 22(7), 4985-5016. https://doi.org/10.5194/acp-22-4985-2022

Gao, K., & Kanji, Z. A. (2022). Impacts of Cloud-Processing on Ice Nucleation of Soot Particles Internally Mixed With Sulfate and Organics. *Journal of Geophysical Research: Atmospheres*, *127*(22), e2022JD037146. https://doi.org/https://doi.org/10.1029/2022JD037146

Gao, K., Koch, H.-C., Zhou, C.-W., & Kanji, Z. A. (2022). The dependence of soot particle ice nucleation ability on its volatile content [10.1039/D2EM00158F]. *Environmental Science: Processes & Impacts*. https://doi.org/10.1039/D2EM00158F

Green, D. W., & Perry, R. H. (2007). *Perry's Chemical Engineers' Handbook, Eighth Edition*. McGraw Hill LLC. https://books.google.ch/books?id=tH7IVcA-MX0C

Huang, C.-H., & Vander Wal, R. L. (2013). Effect of Soot Structure Evolution from Commercial Jet Engine Burning Petroleum Based JP-8 and Synthetic HRJ and FT Fuels. *Energy & Fuels*, *27*(8), 4946-4958. https://doi.org/10.1021/ef400576c

IATA. (2023). *SAF Deployment POLICY*. https://www-prod.iata.org/contentassets/d13875e9ed784f75bac90f000760e998/saf-policy-2023.pdf

Ickes, L., Welti, A., Hoose, C., & Lohmann, U. (2015). Classical nucleation theory of homogeneous freezing of water: thermodynamic and kinetic parameters [10.1039/C4CP04184D]. *Physical Chemistry Chemical Physics*, *17*(8), 5514-5537. https://doi.org/10.1039/C4CP04184D

Jalilehvand, F. (2006). Sulfur: not a "silent" element any more. *Chem Soc Rev*, *35*(12), 1256-1268. https://doi.org/10.1039/b417595f

Jing, L., El-Houjeiri, H. M., Monfort, J.-C., Littlefield, J., Al-Qahtani, A., Dixit, Y., Speth, R. L., Brandt, A. R., Masnadi, M. S., MacLean, H. L., Peltier, W., Gordon, D., & Bergerson, J. A. (2022). Understanding variability in petroleum jet fuel life cycle greenhouse gas emissions to inform aviation decarbonization. *Nature Communications*, *13*(1), 7853. https://doi.org/10.1038/s41467-022-35392-1

Kärcher, B., Busen, R., Petzold, A., Schröder, F. P., Schumann, U., & Jensen, E. J. (1998). Physicochemistry of aircraft-generated liquid aerosols, soot, and ice particles: 2. Comparison with observations and sensitivity studies. *Journal of Geophysical Research: Atmospheres*, *103*(D14), 17129-17147. https://doi.org/https://doi.org/10.1029/98JD01045

Kärcher, B., Möhler, O., DeMott, P. J., Pechtl, S., & Yu, F. (2007). Insights into the role of soot aerosols in cirrus cloud formation. *Atmospheric Chemistry and Physics*, *7*(16), 4203-4227. https://doi.org/10.5194/acp-7-4203-2007

Lee, D. S., Fahey, D. W., Skowron, A., Allen, M. R., Burkhardt, U., Chen, Q., Doherty, S. J., Freeman, S., Forster, P. M., Fuglestvedt, J., Gettelman, A., De León, R. R., Lim, L. L., Lund, M. T., Millar, R. J., Owen, B., Penner, J. E., Pitari, G., Prather, M. J., . . . Wilcox, L. J. (2021). The contribution of global aviation to anthropogenic climate forcing for 2000 to 2018. *Atmospheric Environment*, *244*, 117834. https://doi.org/https://doi.org/10.1016/j.atmosenv.2020.117834

Lee, D. S., Pitari, G., Grewe, V., Gierens, K., Penner, J. E., Petzold, A., Prather, M. J., Schumann, U., Bais, A., Berntsen, T., Iachetti, D., Lim, L. L., & Sausen, R. (2010). Transport impacts on atmosphere and climate: Aviation. *Atmospheric Environment*, *44*(37), 4678-4734. https://doi.org/https://doi.org/10.1016/j.atmosenv.2009.06.005

Liati, A., Schreiber, D., Alpert, P. A., Liao, Y., Brem, B. T., Corral Arroyo, P., Hu, J., Jonsdottir, H. R., Ammann, M., & Dimopoulos Eggenschwiler, P. (2019). Aircraft soot from conventional fuels and biofuels during ground idle and climb-out conditions: Electron microscopy and X-ray micro-spectroscopy. *Environmental Pollution*, *247*, 658-667. https://doi.org/https://doi.org/10.1016/j.envpol.2019.01.078

Lobo, P., Condevaux, J., Yu, Z., Kuhlmann, J., Hagen, D. E., Miake-Lye, R. C., Whitefield, P. D., & Raper, D. W. (2016). Demonstration of a Regulatory Method for Aircraft Engine Nonvolatile PM Emissions Measurements with Conventional and Isoparaffinic Kerosene fuels. *Energy & Fuels*, *30*(9), 7770-7777. https://doi.org/10.1021/acs.energyfuels.6b01581

Lobo, P., Durdina, L., Smallwood, G. J., Rindlisbacher, T., Siegerist, F., Black, E. A., Yu, Z., Mensah, A. A., Hagen, D. E., Miake-Lye, R. C., Thomson, K. A., Brem, B. T., Corbin, J. C., Abegglen, M., Sierau, B., Whitefield, P. D., & Wang, J. (2015). Measurement of Aircraft Engine Non-Volatile PM Emissions: Results of the Aviation-Particle Regulatory Instrumentation Demonstration Experiment (A-PRIDE) 4 Campaign. *Aerosol Science and Technology*, *49*(7), 472-484. https://doi.org/10.1080/02786826.2015.1047012

Mahrt, F., Kilchhofer, K., Marcolli, C., Grönquist, P., David, R. O., Rösch, M., Lohmann, U., & Kanji, Z. A. (2020). The Impact of Cloud Processing on the Ice Nucleation Abilities of Soot Particles at Cirrus Temperatures. *Journal of Geophysical Research: Atmospheres*, *125*(3). https://doi.org/10.1029/2019jd030922

Mahrt, F., Marcolli, C., David, R. O., Grönquist, P., Barthazy Meier, E. J., Lohmann, U., & Kanji, Z. A. (2018). Ice nucleation abilities of soot particles determined with the Horizontal Ice Nucleation Chamber. *Atmospheric Chemistry and Physics*, *18*(18), 13363-13392. https://doi.org/10.5194/acp-18-13363-2018

Marcolli, C., Mahrt, F., & Kärcher, B. (2021). Soot PCF: pore condensation and freezing framework for soot aggregates. *Atmospheric Chemistry and Physics*, *21*(10), 7791-7843. https://doi.org/10.5194/acp-21-7791-2021

Masiol, M., & Harrison, R. M. (2014). Aircraft engine exhaust emissions and other airport-related contributions to ambient air pollution: A review. *Atmospheric Environment*, *95*, 409-455. https://doi.org/https://doi.org/10.1016/j.atmosenv.2014.05.070

Miake-Lye R. C., Martinez-Sanchez M., Brown R. C., & E., K. C. (1993). Plume and wake dynamics, mixing, and chemistry behind a high speed civil transport aircraft. *Journal of Aircraft*, *30*(4), 467-479. https://doi.org/10.2514/3.46368

Moore, R. H., Thornhill, K. L., Weinzierl, B., Sauer, D., D'Ascoli, E., Kim, J., Lichtenstern, M., Scheibe, M., Beaton, B., Beyersdorf, A. J., Barrick, J., Bulzan, D., Corr, C. A., Crosbie, E., Jurkat, T., Martin, R., Riddick, D., Shook, M., Slover, G., . . . Anderson, B.

E. (2017). Biofuel blending reduces particle emissions from aircraft engines at cruise conditions. *Nature*, *543*(7645), 411-415. https://doi.org/10.1038/nature21420

Murray, B. J., Wilson, T. W., Dobbie, S., Cui, Z., Al-Jumur, S. M. R. K., Möhler, O., Schnaiter, M., Wagner, R., Benz, S., Niemand, M., Saathoff, H., Ebert, V., Wagner, S., & Kärcher, B. (2010). Heterogeneous nucleation of ice particles on glassy aerosols under cirrus conditions. *Nature Geoscience*, *3*(4), 233-237. https://doi.org/10.1038/ngeo817

Onasch, T. B., Jayne, J. T., Herndon, S., Worsnop, D. R., Miake-Lye, R. C., Mortimer, I. P., & Anderson, B. E. (2009). Chemical Properties of Aircraft Engine Particulate Exhaust Emissions. *Journal of Propulsion and Power*, *25*(5), 1121-1137. https://doi.org/10.2514/1.36371

Parent, P., Laffon, C., Marhaba, I., Ferry, D., Regier, T. Z., Ortega, I. K., Chazallon, B., Carpentier, Y., & Focsa, C. (2016). Nanoscale characterization of aircraft soot: A high-resolution transmission electron microscopy, Raman spectroscopy, X-ray photoelectron and near-edge X-ray absorption spectroscopy study. *Carbon*, *101*, 86-100. https://doi.org/https://doi.org/10.1016/j.carbon.2016.01.040

Petzold, A., Döpelheuer, A., Brock, C. A., & F., S. (1999). In situ observations and model calculations of black carbon emission by aircraft at cruise altitude. *Journal of Geophysical Research: Atmospheres*, *104*(D18), 22171-22181. https://doi.org/https://doi.org/10.1029/1999JD900460

Petzold, A., Strom, J., Ohlsson, S., & Schroder, F. P. (1998). Elemental composition and morphology of ice-crystal residual particles in cirrus clouds and contrails. *Atmospheric Research*, *49*(1), 21-34. https://doi.org/Doi 10.1016/S0169-8095(97)00083-5

Pires, A. P. P., Han, Y., Kramlich, J., & Garcia-Perez, M. (2018). Chemical composition and fuel properties of alternative jet fuels. *BioResources*, *13(2)*, 2632–2657. https://doi.org/https://doi.org/10.15376/biores.13.2.2632-2657

Popovicheva, O. B., Persiantseva, N. M., Lukhovitskaya, E. E., Shonija, N. K., Zubareva, N. A., Demirdjian, B., Ferry, D., & Suzanne, J. (2004). Aircraft engine soot as contrail nuclei. *Geophysical Research Letters*, *31*(11). https://doi.org/https://doi.org/10.1029/2003GL018888

Schumann, U., Arnold, F., Busen, R., Curtius, J., Kärcher, B., Kiendler, A., Petzold, A., Schlager, H., Schröder, F., & Wohlfrom, K.-H. (2002). Influence of fuel sulfur on the composition of aircraft exhaust plumes: The experiments SULFUR 1–7. *Journal of Geophysical Research: Atmospheres*, *107*(D15), AAC 2-1-AAC 2-27. https://doi.org/https://doi.org/10.1029/2001JD000813

Starik, A. M. (2007). Gaseous and Particulate Emissions with Jet Engine Exhaust and Atmospheric Pollution. *Advances on Propulsion Technology for High-Speed Aircraft*, *Paper 15*( Educational Notes RTO-EN-AVT-150), pp. 15-11 – 15-22.

Twohy, C. H., & Gandrud, B. W. (1998). Electron microscope analysis of residual particles from aircraft contrails. *Geophysical Research Letters*, *25*(9), 1359-1362. https://doi.org/10.1029/97gl03162

Ungeheuer, F., Caudillo, L., Ditas, F., Simon, M., van Pinxteren, D., Kılıç, D., Rose, D., Jacobi, S., Kürten, A., Curtius, J., & Vogel, A. L. (2022). Nucleation of jet engine oil vapours is a large source of aviation-related ultrafine particles. *Communications Earth & Environment*, *3*(1), 319. https://doi.org/10.1038/s43247-022-00653-w

Vander Wal, R., Singh, M., Gharpure, A., Choi, C., Lobo, P., & Smallwood, G. (2022). Turbulence impacts upon nvPM primary particle size. *Aerosol Science and Technology*, *56*(10), 893-905. https://doi.org/10.1080/02786826.2022.2104154

Virtanen, A., Joutsensaari, J., Koop, T., Kannosto, J., Yli-Pirilä, P., Leskinen, J., Mäkelä, J. M., Holopainen, J. K., Pöschl, U., Kulmala, M., Worsnop, D. R., & Laaksonen, A. (2010). An amorphous solid state of biogenic secondary organic aerosol particles. *Nature*, *467*(7317), 824-827. https://doi.org/10.1038/nature09455

Wahl, C., & Aigner, M. (2003). Aircraft Gas Turbine Soot Emission Tests Under Technical Relevant Conditions in an Altitude Test Facility and Validation of Soot Measurement Technique. ASME Turbo Expo 2003, collocated with the 2003 International Joint Power Generation Conference,

Wong, H.-W., Jun, M., Peck, J., Waitz, I. A., & Miake-Lye, R. C. (2014). Detailed Microphysical Modeling of the Formation of Organic and Sulfuric Acid Coatings on Aircraft Emitted Soot Particles in the Near Field. *Aerosol Science and Technology*, *48*(9), 981-995. https://doi.org/10.1080/02786826.2014.953243

Yu, Z., Liscinsky, D. S., True, B., Peck, J., Jennings, A. C., Wong, H.-W., Jun, M., Franklin, J., Herndon, S. C., Waitz, I. A., & Miake-Lye, R. C. (2014). Uptake Coefficients of Some Volatile Organic Compounds by Soot and Their Application in Understanding Particulate Matter Evolution in Aircraft Engine Exhaust Plumes. *Journal of Engineering for Gas Turbines and Power*, *136*(12). https://doi.org/10.1115/1.4027707

Zelenay, V., Ammann, M., Křepelová, A., Birrer, M., Tzvetkov, G., Vernooij, M. G. C., Raabe, J., & Huthwelker, T. (2011). Direct observation of water uptake and release in individual submicrometer sized ammonium sulfate and ammonium sulfate/adipic acid particles using X-ray microspectroscopy. *Journal of Aerosol Science*, *42*(1), 38-51. https://doi.org/https://doi.org/10.1016/j.jaerosci.2010.11.001

Zhang, C., Zhang, Y., Wolf, M. J., Nichman, L., Shen, C., Onasch, T. B., Chen, L., & Cziczo, D. J. (2020). The effects of morphology, mobility size, and secondary organic aerosol (SOA) material coating on the ice nucleation activity of black carbon in the cirrus regime. *Atmospheric Chemistry and Physics*, *20*(22), 13957-13984. https://doi.org/10.5194/acp-20-13957-2020

---

## Author Comment (AC2)

Response to egusphere-2023-2441 reviews for RC2

**Referee comments are marked in black bold and are numbered as R2Cx with x the comment number.** Author (AC) responses are marked in black directly below the comments. The original text from the manuscript is repeated in blue and corrected text in revised manuscript is typed in red. Previous line numbers given in "()" following the updated line numbers.

**Testa et al. comprehensively analyze the aviation engine-emitted soot aerosol particles to understand their ice nucleation ability at cirrus cloud temperatures. Overall, the experiments were well-designed, and the results support their conclusions. Moreover, this study can fill the knowledge gap in the indirect climate effects of soot emitted from aviation engines, which might be an unignore source of soot at high altitudes. I only have some minor comments and questions that can help improve the manuscript. Thus, I recommend publishing it with minor revisions. Please see my comments below:**

We thank Reviewer 2 for their comments and respond to questions and concerns individually below.

**Minor comments:**

**R2C1: Could the authors comment on how high-altitude ambient conditions might affect the results since the experiments were conducted at the ground level?**

The ice nucleation experiments were designed to mimic the atmospheric conditions, i.e., $T$ and $RH_w$ relevant to ice nucleation, and represent by definition high-altitude conditions. The presence of high-altitude background aerosols (dust, haze droplets) would likely reduce the aviation soot AF due to more water vapor competition; this has been explored with air parcel model in Kärcher et al. (2021) and Kärcher et al. (2023) and the quantification of this effect at global scale is planned for future modeling paper using the results of this work.

Then, as mentioned in R1C3 and R1C4, the morphology of the soot primary particles (their size and overlap) is fixed in the combustor. Temperature and pressure in the combustor are high (~2000 K and tens of bar, respectively, Dakhel et al., 2007; Starik, 2007) and largely driven by the engine design and thrust, hence uncorrelated to the ambient conditions. Yet, due to coagulation in the aerosol reservoir, aggregates sampled in our study are larger and lacier (open-branched structure) than uncoagulated particles, such as for high-altitude aviation soot (as discussed in the atmospheric implication section and above). Besides, we acknowledge that the conditions experienced by the soot aggregates before entering the cloud chamber (HINC) differ from high-altitude conditions. In our aerosol reservoir, the temperature and pressure are higher, no nucleation mode particle ($H_2SO_4$ and organic droplets [e.g., oil droplets]) is allowed to form and no dilution with ambient air takes place. As discussed in the paper's Atmospheric implication section, these factors/processes would likely affect the soot mixing state, presumably decreasing the coating over the soot particles (Kärcher et al., 2007; Onasch et al., 2009; Peck et al., 2014; Timko et al., 2013; Wong et al., 2014; Wong et al., 2008) for soot sampled in our study.

**R2C2: I am curious to see the morphology change with and without CS based on the SMPS and Tandem DMA-CPMA measurements.**

Measurements of size change with and without CS with SMPS have been conducted in a companion study (Testa et al., 2024; their Figure 4; simplified in Figure 1 at the end of this document). Those measurements

were conducted at the same maintenance and testing aircraft engine facility and with a similar set up (but focusing on another research question and for different engine types). SMPS measurements from that study (Figure 1 at the end of this document) show that CS-soot aggregate sizes are comparable (within measurement uncertainty) to unCS-soot sizes and hence corroborate the TEM analysis conducted in the present study (Fig. F1).

The analysis of the TEM images of unCS- and CS-soot (Fig. F1) show no change in morphology, and hence we expect fractal dimension of CS-soot to be similar to unCS-soot. Yet, due to the limited time of aerosol sampling from our aerosol reservoir, measurements of CS aggregate mass were conducted only at 200 and 400 nm (as opposed to unCS-soot for which measurements have been conducted at several size points), hence no morphology information (e.g., fractal dimension) can be extracted from the mass measurements due to having only two data points.

**R2C3: Do you expect any physical (e.g., partition on the soot and cause compression) or chemical reaction (oxidation) to happen inside the tank?**

We expect oxidation to be considerably inhibited once exiting the engine combustor (Dakhel et al., 2007) due to the low temperature in the lines and in the aerosol reservoir. However, condensation and evaporation of exhaust gas onto/from the soot particles can occur in the tank due to the drop in temperature from the line to the tank (433 K to 298 K) and the dilution of tank gas with air during the ice nucleation experiment. These changes in particle mixing state were however not characterized, however we acknowledged in the manuscript (see R2C1) that high-altitude soot mixing state likely differs from the soot sampled in our study.

**R2C4: Please note that C, N, and O are semiquantitative in EDX. Moreover, some C and O signal might come from substrates.**

We agree with the reviewer that EDX is only semiquantitative. EDX is nonetheless useful for comparing element concentration of different samples collected on similar grids with the same microscope, as done in this study.

In lines 175-177 (167) we have added: "We point out that although EDX is semiquantitative, it is nonetheless useful for comparing elemental concentrations of different samples collected on similar grids with the same microscope, as done in this study. We further note that sulfur might get vaporized […]"

[Figure]

Figure 1: Mode diameter change $\Delta D_m$ upon processing of the particles with the catalytic stripper measured by SMPS for the given engines. This figure is a modified version of Fig. 4 from Testa et al. (2024)

Dakhel, P. M., Lukachko, S. P., Waitz, I. A., Miake-Lye, R. C., & Brown, R. C. (2007). Postcombustion Evolution of Soot Properties in an Aircraft Engine. *Journal of Propulsion and Power*, *23*(5), 942-948. https://doi.org/10.2514/1.26738

Kärcher, B., Mahrt, F., & Marcolli, C. (2021). Process-oriented analysis of aircraft soot-cirrus interactions constrains the climate impact of aviation. *Communications Earth & Environment*, *2*(1), 113. https://doi.org/10.1038/s43247-021-00175-x

Kärcher, B., Marcolli, C., & Mahrt, F. (2023). The role of mineral dust aerosol particles in aviation soot-cirrus interactions. *Journal of Geophysical Research: Atmospheres*, *n/a*(n/a), e2022JD037881. https://doi.org/https://doi.org/10.1029/2022JD037881

Kärcher, B., Möhler, O., DeMott, P. J., Pechtl, S., & Yu, F. (2007). Insights into the role of soot aerosols in cirrus cloud formation. *Atmospheric Chemistry and Physics*, *7*(16), 4203-4227. https://doi.org/10.5194/acp-7-4203-2007

Onasch, T. B., Jayne, J. T., Herndon, S., Worsnop, D. R., Miake-Lye, R. C., Mortimer, I. P., & Anderson, B. E. (2009). Chemical Properties of Aircraft Engine Particulate Exhaust Emissions. *Journal of Propulsion and Power*, *25*(5), 1121-1137. https://doi.org/10.2514/1.36371

Peck, J., Yu, Z., Wong, H.-W., Miake-Lye, R., Liscinsky, D., Jennings, A., & True, B. (2014). Experimental and Numerical Studies of Sulfate and Organic Condensation on Aircraft Engine Soot. ASME Turbo Expo 2014: Turbine Technical Conference and Exposition,

Starik, A. M. (2007). Gaseous and Particulate Emissions with Jet Engine Exhaust and Atmospheric Pollution. *Advances on Propulsion Technology for High-Speed Aircraft*, *Paper 15*( Educational Notes RTO-EN-AVT-150), pp. 15-11 – 15-22.

Testa, B., Durdina, L., Edebeli, J., Spirig, C., & Kanji, Z. A. (2024). Contrail processed aviation soot aerosol are poor ice nucleating particles at cirrus temperatures. *EGUsphere*, *2024*, 1-22. https://doi.org/10.5194/egusphere-2024-151

Timko, M. T., Fortner, E., Franklin, J., Yu, Z., Wong, H. W., Onasch, T. B., Miake-Lye, R. C., & Herndon, S. C. (2013). Atmospheric Measurements of the Physical Evolution of Aircraft Exhaust Plumes. *Environmental Science & Technology*, *47*(7), 3513-3520. https://doi.org/10.1021/es304349c

Wong, H.-W., Jun, M., Peck, J., Waitz, I. A., & Miake-Lye, R. C. (2014). Detailed Microphysical Modeling of the Formation of Organic and Sulfuric Acid Coatings on Aircraft Emitted Soot Particles in the Near Field. *Aerosol Science and Technology*, *48*(9), 981-995. https://doi.org/10.1080/02786826.2014.953243

Wong, H.-W., Yelvington, P. E., Timko, M. T., Onasch, T. B., Miake-Lye, R. C., Zhang, J., & Waitz, I. A. (2008). Microphysical Modeling of Ground-Level Aircraft-Emitted Aerosol Formation: Roles of Sulfur-Containing Species. *Journal of Propulsion and Power*, *24*(3), 590-602. https://doi.org/10.2514/1.32293

---

## Author Comment (AC3)

Response to egusphere-2023-2441 reviews for RC3

**Referee comments are marked in black bold and are numbered as R3Cx with x the comment number.** Author (AC) responses are marked in black directly below the comments. The original text from the manuscript is repeated in blue and corrected text in revised manuscript is typed in red. Previous line numbers given in "()" following the updated line numbers.

**R3C1: In this study Testa et al. investigates the ice nucleation activity of aviation soot that were sampled directly from the modern in-use commercial aircraft engines. The effects of engine thrust and soot particle size, mixing state, physical and chemical properties were tested in various experiments using a continuous flow diffusion chamber. Their results indicate that the overall ice nucleation abilities of real aviation soot are not so high as previously thought since in the latter case surrogate of aviation soot were used to estimate their ice nucleation activity. The study is very interesting and meaningful for further evaluating ice formation and climate effect associated with aviation. I recommend the paper to be published in ACP with minor revisions. What I suggest is that the authors may consider adding a short comment/discussion about the implication and usage of their experiment results and data for modelling work in the Conclusions.**

We thank Reviewer 3 for their comment on the manuscript. The results from this work are indeed being currently used to estimate soot-cirrus radiative properties based on the ice nucleation results of this work. This will be published in a separate study in the future.

We added the following to the conclusion in lines 575-579 (562): "[…] allow better predictive capability of aviation soot ice nucleation. Results from this study will be applied in future modelling work to update estimates of radiative forcing from aviation soot. For the most representative simulation, we suggest using the ice nucleation properties ([onset] RH for given AF) of our mixed-thrust unCS-soot sample, to closely mimic in situ aviation soot. Given the RH onset of mixed-thrust unCS-soot is close to or at $RH_{hom}$, we expect their effect on cirrus cloud and hence, their radiative forcing to be negligible."